# Towards Robust Pruning: An Adaptive Knowledge-Retention Pruning Strategy for Language Models

**Jianwei Li[1]   Qi Lei[2]   Wei Cheng[3]   Dongkuan Xu[1]**

[1]North Carolina State University, {jli265, dxu27}@ncsu.edu
[2]New York University, ql518@nyu.edu
[3]NEC-Labs, weicheng@nec-labs.com

## Abstract

The pruning objective has recently extended beyond accuracy and sparsity to robustness in language models. Despite this, existing methods struggle to enhance robustness against adversarial attacks when continually increasing model sparsity and require a retraining process. As humans step into the era of large language models, these issues become increasingly prominent. This paper proposes that the robustness of language models is proportional to the extent of pre-trained knowledge they encompass. Accordingly, we introduce a post-training pruning strategy designed to faithfully replicate the embedding space and feature space of dense language models, aiming to conserve more pre-trained knowledge during the pruning process. In this setup, each layer's reconstruction error not only originates from itself but also includes cumulative error from preceding layers, followed by an adaptive rectification. Compared to other state-of-art baselines, our approach demonstrates a superior balance between accuracy, sparsity, robustness, and pruning cost with BERT on datasets SST2, IMDB, and AG-News, marking a significant stride towards robust pruning in language models.

## 1 Introduction

Pruning is a widely recognized compression method employed to decrease the model size and accelerate model inference (Frankle and Carbin, 2018; Chen et al., 2020; Prasanna et al., 2020; Chen et al., 2021). In the age of large language models (Andrew and Gao, 2007; Brown et al., 2020; Chowdhery et al., 2022; OpenAI, 2023; Touvron et al., 2023; Ouyang et al., 2022; Smith et al., 2022), the necessity of pruning has increased because it greatly reduces deployment costs (Frantar and Alistarh, 2023). In addition to the significant computation cost, the robustness of language models has emerged as a crucial factor that demands attention. This is primarily because models need to remain resilient against adversarial attacks, even in challenging real-world circumstances (Tran et al., 2022; Wang et al., 2023). Therefore, exploring robust pruning strategies against adversarial attacks in language models could potentially yield a substantial impact (Xu et al., 2021; Du et al., 2023).

Recent research has extended the pruning of language models beyond accuracy and sparsity, with an emphasis on the trade-off between accuracy, sparsity, robustness and cost (Du et al., 2023; Xu et al., 2021; Liang et al., 2021; Xi et al., 2022). Zheng et al. (2022) propose a joint optimization objective to guide the pruning and adversarial training simultaneously. Their approach views the identified subnetworks as robust tickets, which can be trained as normal and offer enhanced robustness. Despite achieving state-of-the-art results on target datasets, these methods still display vulnerabilities, as evidenced by a significant gap between metrics of clean accuracy [1] and accuracy under attack. Moreover, the performance also rapidly declines when sparsity exceeds a moderate level. Expanding on their work, Xi et al. (2022) propose using robust early-bird tickets to reduce the computational cost from adversarial training. However, they face similar challenges regarding the trade-off between robustness and sparsity. In summary, existing robust pruning works often demonstrate limited sparsity, insufficient robustness, and expensive cost, indicating the ongoing challenge of the balance between accuracy and the other three aspects.

To address this challenge, this paper investigates why language models are susceptible to adversarial attacks. (Wang et al., 2021; Garg and Ramakrishnan, 2020; Jin et al., 2020). Previous studies have indicated that language models frequently capitalize on biases and artifacts inherent in datasets as predictive shortcuts, which impedes reasoning ability and skills to develop advanced semantic comprehension. (Du et al., 2021; Niven and Kao, 2019;

---

[1]accuracy without adversarial attacks

McCoy et al., 2020; Du et al., 2023). This reliance leads to a more severe loss of pre-trained knowledge during the pruning process. Furthermore, the adversarial samples in Natural Language Processing (NLP) are crafted by replacing components of sentences with semantically similar counterparts, thereby retaining high semantic similarity in the entire sentence (Li et al., 2020a; Ren et al., 2019; Jin et al., 2020). In this way, language models that depend on spurious features from particular words can not defend against adversarial attacks constructed by replacing those words with semantically similar alternatives. To put it more plainly, this primarily stems from the fact that, without pre-trained knowledge, the sparse language model treats the substitute word simply as an integer identifier. Based on the above observation, we explore the following questions in this paper:

**Question 1.** *What is the core to defend against adversarial attacks for sparse language models?*

This paper proposes that the robustness of sparse language models is directly proportional to the amount of pre-trained knowledge retained after pruning. Intuitively, the robustness of a sparse language model is fundamentally tied to its capability to distill advanced semantic features from input sentences. This capability is largely established during the pre-training phase of dense language models, emphasizing the pivotal role of acquired semantic knowledge. The extensive experiments well support our statement.

**Question 2.** *How can we efficiently prevent the loss of pre-trained knowledge in pruning to preserve or even enhance robustness?*

Previous research has demonstrated that pruning exacerbates the model's dependency on spurious features (Xu et al., 2021; Du et al., 2023). We further confirm that traditional pruning methods lead to a considerable loss of pre-trained knowledge and poor robustness. To prevent the above things, we propose a pruning approach that minimizes damage to the embedding space and feature space of dense language models, striving to replicate the features in each layer completely. Specifically, for each layer, we iteratively eliminate a single weight at a time and counterbalance the loss by updating the remaining weights based on the Hessian Matrix. In this setup, the reconstruction error at each layer arises not only from its own layer but also incorporates the accumulated error from preceding layers. This is achieved by adaptively updating

the pruning-dependent information in accordance with the sparse output generated by previous layers. Concurrently, there's an ongoing effort to correct these errors collectively. Moreover, our method, being a post-training approach, is cost-effective for current language models, as it circumvents rigorous retraining processes. Extensive experiments show that our approach achieves a better trade-off between accuracy, sparsity, robustness, and pruning cost in SST2, AGNews, and IMDB compared with other state-of-art methods.

## 2 Related Work

**Textual Adversarial Attacks and Defense.** Textual adversarial attacks pose a significant challenge to the robustness of language models. These attacks, formulated by carefully altering certain segments of sentences with semantically similar counterparts, aim to fool language models (Jin et al., 2020; Li et al., 2020a). To enhance the robustness of language models and defend against adversarial attacks, a range of potent defensive strategies, such as adversarial training, has been proposed. (Madry et al., 2017; Zhu et al., 2019; Li and Qiu, 2021). Different from their research, which focuses on dense models, we explore the robustness in the context of language model pruning.

**Robust Model Pruning.** Prior studies indicate that sparse models tend to underperform in Compression Identified Examples (CIE), suggesting that the pruning process exacerbates the inherent algorithmic biases hidden within the datasets (Hooker et al., 2020). In Computer Vision (CV), simultaneous optimization of model pruning and adversarial training has been advocated as an effective solution to this issue (Gui et al., 2019; Ye et al., 2019; Sehwag et al., 2020; Vemparala et al., 2021). In NLP, Du et al. (2023) propose to prevent model overfitting on easy samples by leveraging sample difficulty in the context of pruning. Concurrently, Xu et al. (2021) suggest the generation of robust subnetworks through Knowledge Distillation and Post-training Quantization. Taking a different approach, Liang et al. (2021) strive to enhance model generalizability by extracting the super tickets, while Zheng et al. (2022) and Xi et al. (2022) seek to identify robust tickets. Despite recent advancements, achieving enhanced robustness alongside increased sparsity remains a challenge. This paper significantly promotes a better trade-off among accuracy, robustness, sparsity, and pruning cost.

## 3  Preliminary

### 3.1  Shortcut Learning and Mitigation

Recent studies provide evidence that language models are inclined to capitalize on inherent biases and spurious features present in datasets, using these as convenient predictive shortcuts (Niven and Kao, 2019; Du et al., 2021; McCoy et al., 2020). This tendency impedes the development of more advanced semantic understanding and reasoning capacity necessary for NLU tasks. Various preliminary studies have begun to address this bias issue, such as adversarial training and posterior regularization (Stacey et al., 2020; Chen et al., 2021). From a unique perspective, we let language models against adversarial attacks by mitigating this shortcut issue through *weight averaging*. This method will be elaborated further in Section 4.2.

### 3.2  Pruning with Hessian Matrix

Drawing inspiration from (LeCun et al., 1989; Hassibi et al., 1993), previous study has provided mathematical formulations for effectively eliminating a single weight from a layer and updating the remaining weights to correct the resulting error according to the information from Hessian Matrix (Frantar and Alistarh, 2022). The equations are presented below:

$$w_p = \underset{w_p}{\operatorname{argmin}} \frac{w_p^2}{[H^{-1}]_{pp}}$$
$$w_r- = \frac{w_p}{[H^{-1}]_{pp}} \cdot H_{:,p}^{-1} \tag{1}$$

where $H$ is the Hessian Matrix, $w_p$ represents the single weight that will be pruned, while $w_r$ denotes the remaining weights that will be updated. The notation $[H^{-1}]pp$ refers to the $p_{th}$ diagonal entry of the inverse Hessian Matrix, and $H_{:,p}^{-1}$ represents its $p_{th}$ column. However, the inversion of the Hessian Matrix requires updates at each weight removal, which is exceedingly costly. Frantar and Alistarh (2022) observes that Hessian values across different weight matrix rows are independent, as a single weight removal only impacts its respective row output. Accordingly, they simplify the calculation of Hessian Matrix $H$ and leverage the Gaussian elimination technique to accelerate the update of $H^{-1}$, as described mathematically below:

$$H = XX^T$$
$$H_{-p}^{-1} = (H^{-1} - \frac{1}{[H^{-1}]_{pp}} H_{:,p}^{-1} H_{p,:}^{-1})_{-p} \tag{2}$$

Here, $-p$ denotes the removal action of a single weight at index $p$. A more detailed explanation can be found in the Appendix.

## 4  Methodology

This section proposes a pruning method for language models that can better balance accuracy, sparsity, robustness, and pruning cost. Figure 1 depicts the architecture of this method.

### 4.1  Rethink Robust Model Pruning

Given that the predominant challenge in robust pruning primarily centers on robustness and pruning cost, we mainly focus on these two aspects in this paper. To enhance the robustness, we explore the root cause of the poor performance of sparse language models under adversarial attacks. We note that adversarial samples are often crafted by replacing certain words in the sentence with semantically similar substitutes. Thus it is essential to ensure that the representation of the original words and their substitutes remain similar in the embedding space and feature space even after pruning. Based on the above observation, we propose to maintain a highly close alignment between the sparse and dense language models. In other words, robust pruning is supposed to seek sparse parameters $\hat{W}_l$ that minimize the discrepancy between the outputs of dense and sparse layers. The problem can be formally expressed as follows:

$$\operatorname{argmin}_{\hat{W}_l} E_{X_l} \mathcal{L}(f_l(X_l, W_l), f_l(X_l, \hat{W}_l))$$
$$\text{s.t. } \|\hat{W}_l\|_0 \leq k \tag{3}$$

Here, each layer of language models is represented by a mathematical function $f_l(W_l, X_l)$, and $X_l$ denotes inputs, $k$ designates the total number of weights that remain non-zero after the pruning process. Predominantly, the Mean Squared Error (MSE) is usually employed to measure the pruning error of each layer. Therefore, the preceding problem can be further reformulated using the MSE, as expressed in the subsequent equation:

$$\operatorname{argmin}_{\hat{W}_l} \|W_l X_l - \hat{W}_l X_l\|^2 \text{ s.t. } \|\hat{W}_l\|_0 \leq k \tag{4}$$

To reduce the pruning cost, we adopt a post-training setting in our strategy. Specifically, we only utilize a small subset of data to calibrate the weights and generate sparse substitutes to replace them. In summary, our pruning method does not need a rigorous retraining process.

### 4.2  Weight Averaging for Robust Dense Model

We also realize that language models may rely on surface-level or spurious features in the data

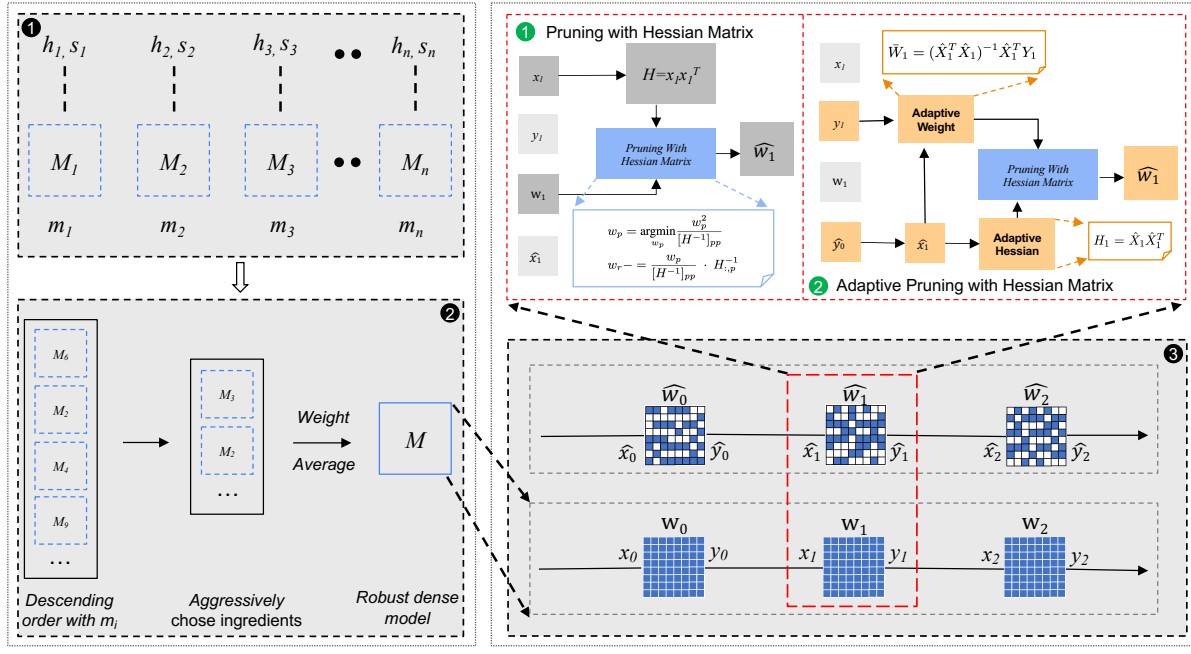

Figure 1: Architecture of Main Strategy. **A:** First, we generate a robust and dense language model in two steps: ❶ we fine-tune the pre-trained weight with various hyperparameters and settings, resulting in multiple models with different knowledge; ❷ we then employ a greedy algorithm to only average the weights of models that contribute to the final performance. **B:** Second, ❸ we apply our adaptive pruning method to generate robust and sparse language models in a layer-wise setting. Specifically, we optimize the ❶ original independent pruning process of each layer to ❷ an adaptive way. This requires subsequent layers to update the Hessian Matrix and the optimal dense weight according to the sparse outputs of preceding layers, thereby inheriting and correcting the accumulated error together.

rather than capturing sophisticated semantic features. Thus, when sparse language models fail to defend against adversarial attacks, it becomes challenging to determine whether the failure stems from the pruning methods or inherent issues within the dense model. We circumvents this risk by constructing a robust and dense model before pruning.

Inspired by Croce et al. (2023) and Wortsman et al. (2022), we generate a robust language model via *weight averaging*. The key idea is to train multiple models with different hyperparameters and settings, allowing each model to capture distinct nuances of the data and generalize in diverse ways. By averaging their weights, we can create a robust model that benefits from collective knowledge. Specifically, we order these models in descending order based on the accuracy under attack. Then, we selectively average the weights that contribute to the final robustness. Finally, we obtain a robust and dense model as the foundation of subsequent operations. This approach ensures that any detected vulnerabilities in sparse language models result from the pruning process, eliminating the possibility of them arising from spurious features. More

details can be found in Algorithm 3.

### 4.3 Ada-Pruning for Robust Sparse Model

#### 4.3.1 Notation

To accurately replicate the dense model's behavior regarding embedding space and feature space of each layer, we use the method described in Section 3.2 as the backbone. However, its layer-wise setting, which treats each layer as an independent pruning problem, introduces limitations in realizing a globally optimal solution. To elaborate, let's consider a single layer as an example in the following sections. We'll use $X_l$, $W_l$, and $Y_l$ to represent the input, weight, and output of the layer, respectively, with the subscript $l$ indicating $l_{th}$ layer. The use of a hat, as seen in $\hat{X}_l$, $\hat{W}_l$, or $\hat{Y}_l$, represents the input, weight, or output within a sparse context.

#### 4.3.2 Adaptive Hessian Matrix

After completing the pruning of the $l_{th}$ layer, a certain amount of error stemming from the sparse matrix operation inevitably arises. No matter how minor this error might be, it's important to realize that the output of this layer, denoted as $\hat{Y}_l$, influ-

**Algorithm 1** Prune linear layers $\{l_1..l_n\}$ of BERT with target sparsity $s$ and calibration data $X$

---

**Require:** Collect original $X, W, Y$ for $l$
1: **procedure** LAYERWISE PRUNING($\{l_1..l_n\}$)
2:     **for** $i \leftarrow 1$ to $n$ **do**
3:         $W_i, X_i, Y_i \leftarrow l_i$
4: - - - - - - - - - - - - - - - - - - - - - - - - - - - - - - -
5:         *# Adaptive update*
6:         $H_i^{-1} \leftarrow (X_i X_i^T)^{-1}$
7:         **if** $i \neq 0$ **then**
8:             $W_i \leftarrow H_i^{-1} X_i^T Y_i$
9:         **end if**
10: - - - - - - - - - - - - - - - - - - - - - - - - - - - - - -
11:         *# Pruning with Hessian Matrix*
12:         $d_{in} \leftarrow$ input dimension
13:         $k \leftarrow$ int $(d_{in} \cdot s)$
14:         **for** $j \leftarrow 1$ to $k$ **do**           ▷ parallel in rows
15:             $p \leftarrow argmin_{p \in d_{in}} \frac{1}{[H_i^{-1}]_{pp}} \cdot [W_i]_p^2$
16:             $W_i \leftarrow W_i - [H_i]_{:,p}^{-1} \frac{1}{[H_i^{-1}]_{pp}} \cdot [W_i]p$
17:             $\text{tmp} \leftarrow [H_i]_{:,p}^{-1}[H_i]_{p,:}^{-1}$
18:             $H_i^{-1} \leftarrow H_i^{-1} - \frac{1}{[H_i^{-1}]_{pp}} tmp$
19:             $W_i \leftarrow W_i$ remove $[W_i]_p$
20:         **end for**
21: - - - - - - - - - - - - - - - - - - - - - - - - - - - - - -
22:         *# Adaptive update*
23:         $Y_i \leftarrow W_i X_i$
24:         $X_{i+1} \leftarrow$ post-process($Y_i$)
25:     **end for**
26:     **return** $\{W_i..W_n\}$
27: **end procedure**

---

ences the input of the subsequent layer, denoted as $\hat{X}_{l+1}$. As a result, the initial Hessian Matrix for the $(l+1)_{th}$ layer, defined as $H_{l+1} = X_{l+1}X_{l+1}^T$, becomes outdated. Thus it's crucial to recalculate the Hessian Matrix to obtain more precise pruning-dependent information. We suggest adaptively updating the Hessian Matrix for the subsequent layer after pruning the preceding layers.

### 4.3.3 Adaptive Dense Weight

We also note that the loss generated by removing a single weight depends on the current weight $W_l$ from corresponding layer, as derived from Equation 1. However, an inevitable fact is that the original dense weight $W_l$ is not optimal for the expected dense output $Y_l$ after pruning the preceding layers $(\hat{0}_{th} \ldots (l \overset{\hat{}}{-} 1)_{th})$. Given that the input $X_l$ has been altered to $\hat{X}_l$ due to the accumulated error, it would be suboptimal to continue using the original weight $W_l$ to calculate the pruning loss for the current layer. To be more clear, the result of $\hat{X}_l W_l$ could substantially deviate from the original output $Y_l$. This is incompatible with our goal of producing an output $\hat{Y}_l$ identical to the original $Y_l$ in the pruning process. Thus, it's essential to update the dense weight so that $\hat{X}_l \bar{W}_l$ can approximates the original

output $Y_l$ more closely. Here, $\bar{W}_l$ denotes the updated dense weight, and we design the following equations to derive $\bar{W}_l$:

$$\bar{W}_l = (\hat{X}_l^T \hat{X}_l)^{-1} \hat{X}_l^T Y_l \tag{5}$$

where $T$ represents the transpose operation, and $-1$ denotes the inverse operation. To ensure that $\hat{X}_l^T \hat{X}_l$ is invertible, we also introduce a regularization term, such as $1e - 4$, to the diagonal entries of the matrix. Finally, we can compute the pruning loss more accurately with the updated weight $\bar{W}_l$.

We also calibrate the optimal weights for non-pruned layers (such as the pooler layer and classification layer in BERT) with Equation 5, aligning the dense layers' output with the altered input. Algorithm 1 provides detailed steps for the code implementation, offering a comprehensive overview of our methodology. We also provide a comprehensive analysis of the computational complexity of our method in the Appendix.

## 5 Experiments

We first compare our method against several baseline methods, assessing accuracy, robustness, sparsity, and cost. Then, an ablation study is performed to elucidate the contributions of each part in our method. Finally, we augment our core findings with additional experiments and analyses to further illuminate our method.

### 5.1 Baselines and Datasets

Consistent with the previous works (Devlin et al., 2018; Du et al., 2023; Xu et al., 2021; Zheng et al., 2022; Xi et al., 2022), **BERT**$_{base}$ serves as the foundational model for all our experiments. We compare our approach with various baselines including:**RobustT** (Zheng et al., 2022), which optimizes the pruning mask and input perturbation simultaneously for robust tickets; **Bag-of-Ticks** (Xu et al., 2021), which improves sparse model robustness via Knowledge Distillation and Post-Training Quantization; **RMC** (Du et al., 2023), a technique preventing sparse language models from overfitting on easy samples using sample difficulty; **SuperTicket** (Liang et al., 2021), which identifies a super mask during pruning to reduce variance while preserving bias. Our evaluation primarily involves three text classification datasets: Internet Movie Database (**IMDB**, Maas et al. 2011), AG News Corpus (**AGNEWS**, Zhang et al. 2016), and Stanford Sentiment Treebank for binary classification (**SST-2**, Socher et al. 2013).

| Methods | #Param | Re-T | SST2 | | | AGNEWS | | | IMDB | | |
|---|---|---|---|---|---|---|---|---|---|---|---|
| | | | Acc | Aua | Asr | Acc | Aua | Asr | Acc | Aua | Asr |
| Fine-tune | 85M | Y | **92.3** | 12.7 | 86.2 | 94.7 | 19.1 | 80.0 | 95.1 | 7.4 | 92.2 |
| FreeLB | 85M | Y | 91.5 | 28.3 | 69.1 | **94.8** | 37.8 | 60.1 | 94.3 | 36.2 | 61.6 |
| Weight Average | 85M | Y | 91.4 | **30.4** | **66.75** | 94.4 | **48.5** | **48.6** | **95.2** | **44.4** | **53.4** |
| *sparsity ≤ 30%* | | | | | | | | | | | |
| SuperTicket | 72M | Y | **93.2** | 14.3 | 84.7 | 94.8 | 9.7 | 89.8 | **95.0** | 17.3 | 81.8 |
| Bag-of-Tricks | 60M | N | 86.3 | 25.7 | 70.3 | 87.3 | 31.8 | 63.6 | 85.4 | 24.6 | 71.2 |
| RMC | 60M | Y | 91.2 | 17.6 | 80.7 | 94.2 | 21.4 | 77.3 | 93.9 | 22.3 | 76.3 |
| RobusT | 60M | Y | 90.8 | 28.9 | 68.2 | **94.9** | 33.4 | 64.8 | 92.1 | 55.7 | 39.5 |
| Ours | 60M | N | 90.2 | **42.3** | **53.1** | 93.8 | **48.6** | **48.2** | 94.6 | **57.3** | **39.4** |
| *sparsity = 50%* | | | | | | | | | | | |
| Bag-of-Tricks | 43M | N | 87.2 | 21.6 | 75.2 | 90.6 | 33.5 | 63.0 | 91.3 | 21.2 | 76.8 |
| RMC | 43M | Y | **90.8** | 9.7 | 89.3 | 94.1 | 21.2 | 77.5 | 94.1 | 14.7 | 84.4 |
| RobusT | 43M | Y | 90.5 | 24.8 | 73.9 | **94.8** | 28.8 | 69.7 | 93.2 | 31.5 | 66.2 |
| Ours | 43M | N | 88.31 | **43.1** | **51.2** | 93.4 | **48.5** | **48.1** | **94.2** | **53.2** | **43.6** |
| *sparsity = 87.5%* | | | | | | | | | | | |
| Bag-of-Tricks | 11M | N | 85.9 | 17.8 | 85.7 | 89.4 | 11.3 | 87.4 | 87.7 | 8.9 | 89.9 |
| RMC | 11M | Y | **86.3** | 3.6 | 95.8 | 92.1 | 4.5 | 95.5 | 91.3 | 11.2 | 87.7 |
| RobusT | 11M | Y | 85.2 | 7.8 | 90.8 | 91.8 | 8.3 | 91.0 | 89.2 | 6.5 | 92.7 |
| Ours | 11M | N | 85.6 | **37.6** | **56.1** | 92.4 | **41.3** | **55.3** | 91.6 | **35.6** | **61.1** |

Table 1: Summary of Adversarial Robustness Assessment on BERT$_{base}$. The entry highlighted with an **orange background** denotes our robust and dense model, which serves as the initialization for a range of robust pruning methods except **RobustT** (RobustT is generated from the pre-trained weight). Obviously, our method consistently outperforms all baselines in terms of the **Aua%** and **Asr%** metrics. Regarding **Acc%**, there is a minor decrease in our method's performance at lower sparsity levels, yet it regains superiority at higher sparsity levels. The highest performance is highlighted in **bold**. The column **Re-T** indicates whether the method necessitates model retraining. Consistent with previous research, we exclude embedding matrices from the calculation of parameter count.

## 5.2 Robustness Evaluation

We assess our model's effectiveness against adversarial attacks using the **TextFooler**, which substitutes crucial words in sentences with semantically similar synonyms (Jin et al., 2020). Following previous works (Zheng et al., 2022; Xi et al., 2022), our evaluations utilize key metrics like Clean Accuracy **Acc%** (accuracy on clean test data), Accuracy Under Attack **Aua%** (accuracy when subjected to adversarial attacks), and Attack Success Rate **Asr%** (ratio of successful text perturbations to total attempts). A robust method is expected to show higher clean accuracy and accuracy under attack coupled with a lower attack success rate. We also evaluate more attack methods in the Appendix.

## 5.3 Implementation Details

To begin with, we employ the technique mentioned in Section 4.2 to generate a robust language model for each dataset. Subsequently, we use our method to prune these robust language models with a small calibration dataset. All experimental results are the average of five trials, each initiated with different

seeds. Furthermore, we assess the performance under three different levels of sparsity: 30%, 50%, and 87.5%. Additional implementation details can be found in Appendix.

## 5.4 Main Result on Robustness Evaluation

Table 1 provides a comprehensive comparison of various robust pruning methods, evaluated across three distinct datasets: SST2, AGNEWS, and IMDB, and under varying degrees of model sparsity. Key observations can be made as follows: **1)** Our strategy even enhances the robustness of language models after pruning. We believe this enhancement stems from the regularization effect of sparse architecture. **2)** Our strategy distinguishes itself by consistently surpassing other methods in the **Aua%** and **Asr%**s, regardless of the dataset or the level of sparsity. These results imply that our strategy effectively maintains robustness during the pruning of language models. **3)** Impressively, our method achieves higher robustness even with fewer parameters compared to several other approaches, which further underscores the effectiveness of our robust pruning method. **4)** Although the **Acc%** of

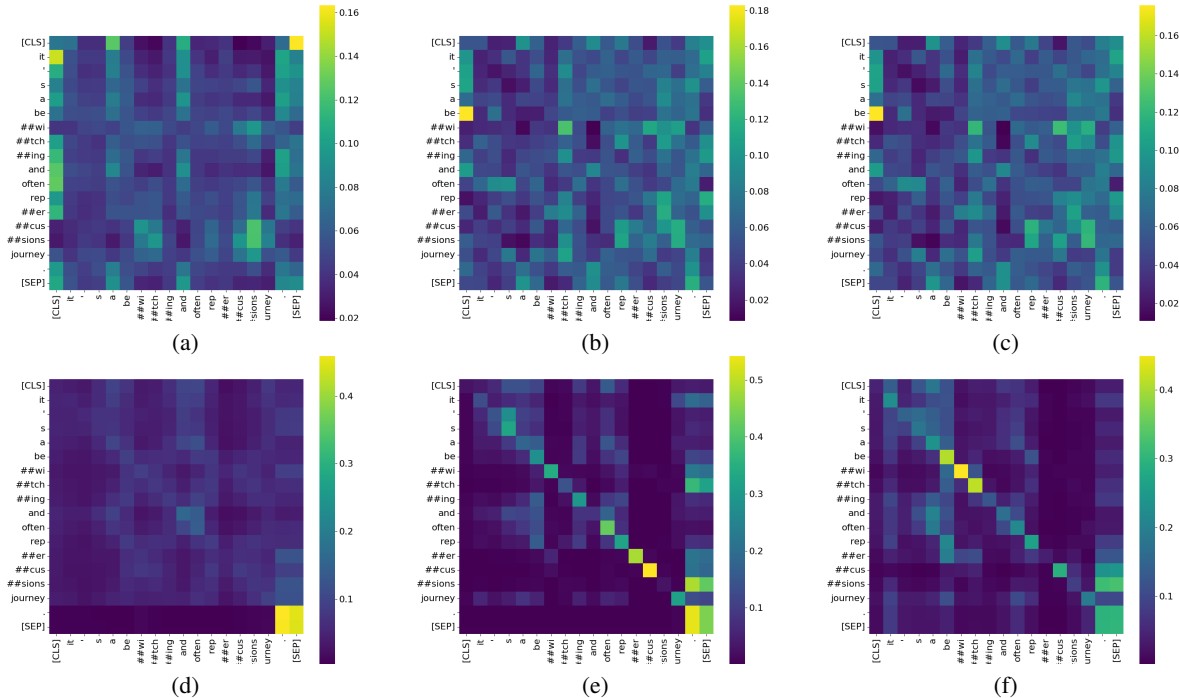

Figure 2: Attention Score Visualisation in BERT$_{base}$. We have selected an adversarial sample ("*it's a bewitching and often repercussions journey.*") from SST2 and visualized the attention scores in the robust and dense model (2b, 2e), the sparse language model generated with IMP+FreeLB (2a, 2d), and the sparse language model created using our method (2c, 2f). Here, Figures 2a, 2b, and 2c depict the attention scores from the first transformer block of BERT$_{Base}$, while Figures 2d, 2e,and 2f show scores from the last transformer block. Evidently, the attention scores produced by our method align more closely with those from the robust and dense model.

| Methods | #Param | ReT | SST2 | | | AGNEWS | | | IMDB | | |
|---|---|---|---|---|---|---|---|---|---|---|---|
| | | | Acc | Aua | Asr | Acc | Aua | Asr | Acc | Aua | Asr |
| Fine-tune | 85M | Y | **92.3** | 12.7 | 86.2 | 94.7 | 19.1 | 80.0 | 95.1 | 7.4 | 92.2 |
| Weight Average | 85M | Y | 91.4 | **30.4** | **66.75** | 94.4 | **48.5** | **48.6** | **95.2** | **44.4** | **53.4** |
| IMP | 43M | Y | **92.6** | 4.8 | 94.8 | **94.9** | 7.1 | 92.5 | 94.1 | 7.7 | 91.8 |
| IMP + FreeLB | 43M | Y | 92.4 | 7.9 | 91.5 | 94.3 | 9.2 | 90.2 | 93.8 | 14.3 | 84.8 |
| LTH | 43M | Y | 91.6 | 2.8 | 96.9 | 93.5 | 10.1 | 89.2 | 93.2 | 4.6 | 95.1 |
| LTH + FreeLB | 43M | Y | 91.7 | 9.8 | 89.3 | 93.2 | 12.3 | 86.8 | 93.1 | 9.5 | 89.8 |
| Ours | 43M | N | 88.31 | **43.1** | **51.2** | 93.4 | **48.5** | **48.1** | 94.2 | **53.2** | **43.6** |

Table 2: Ablation Study with Pruning Methods Replacement. We replace our pruning method with most famous others (**IMP** and **LTH**) supplemented with adversarial training (**FreeLB**). Similarly, the orange entry is used for model initialization. Once again, our method outperforms others in preserving or even enhancing robustness.

our method is generally lower than other baselines at lower sparsity levels, the improvement of robustness (reflected in **Aua%** and **Asr%**) far outweighs the degree of accuracy degradation. **5)** At higher levels of sparsity, our method outperforms other baselines across all metrics. **6)** Our method does not require model retraining, confirming that our approach offers a better trade-off between accuracy, robustness, sparsity, and pruning cost.

Beyond Bert$_{base}$, our methodology was also extended to Bert$_{large}$, a model encompassing 330M parameters. The resulting performance, as presented in Table 3, reaffirms the superiority of our

method when compared to the baselines. Moreover, we explore the effectiveness of our methods within a structured pruning context, and once again, our approach outperforms the state-of-the-art method: **EarlyRobust** (Xi et al., 2022). More details can be found in Appendix.

## 5.5 Ablation Study

To elucidate the contributions of each part of our approach, we conduct an ablation study with the following settings:We replace our pruning technique with methods known as **LTH** and **IMP** (Frankle et al., 2020; Frankle and Carbin, 2018), and supple-

| Methods | #Param | Re-T | SST2 | | | AGNEWS | | | IMDB | | |
|---|---|---|---|---|---|---|---|---|---|---|---|
| | | | Acc | Aua | Asr | Acc | Aua | Asr | Acc | Aua | Asr |
| Weight Average | 309M | Y | 93.5 | 36.4 | 61.1 | 96.2 | 56.5 | 41.3 | 95.9 | 48.4 | 49.6 |
| Bag-of-Tricks | 155M | N | 90.3 | 27.6 | 69.4 | 93.1 | 35.5 | 61.9 | 93.4 | 29.3 | 68.6 |
| RMC | 155M | Y | **92.6** | 14.7 | 84.1 | 95.4 | 19.2 | 79.9 | **95.8** | 16.7 | 82.6 |
| RobusT | 155M | Y | 92.1 | 29.8 | 67.7 | 95.1 | 32.8 | 65.6 | 95.2 | 31.9 | 66.5 |
| Ours | 155M | N | 91.7 | **47.1** | **48.6** | **95.5** | **53.5** | **44.0** | 95.3 | **55.8** | **41.4** |

Table 3: Summary of Adversarial Robustness Assessment on BERT$_{large}$. Similarly, the entry highlighted with an orange background is used for model initialization. Once again, our method consistently outperforms all baselines in terms of the **Aua%** and **Suc%** metrics.

ment them with the additional adversarial training method **FreeLB** (Zhu et al., 2019). The results are presented in Table 2. From the results, we can make the following key observations: 1) Sparse language models generated by traditional pruning methods performs even worse than the vanilla fine-tuned dense model. This highlights the challenges associated with robust pruning. 2) Our approach consistently generates more robust sparse language models than conventional pruning methods, even supplemented with adversarial training methods. 3) We conjecture that the limited effect of adversarial training here stems from the discrete nature of word tokens and the substantial loss of pre-trained knowledge during pruning.

## 5.6 Discussion

In this section, we design additional experiments to illustrate our robust pruning method further.

### 5.6.1 Pretrained Knowledge Detection

To demonstrate the effectiveness of our robust pruning mechanism in preserving pre-trained knowledge, we've chosen adversarial samples that are effectively defended by our method but not by others. We then visualize the attention scores of them in Figure 2. Our method demonstrates superior performance, as evidenced by more reasonable attention scores that align more closely with those from the robust and dense model. In addition, we visualize the distance of sentence representation from sparse language models and their dense counterparts in the feature space. As depicted in Table 4 and Figure 5, our method results in smaller distances between the dense and sparse representations. These findings indicate the superior ability of our robust pruning method to preserve semantic knowledge and maintain cognizance. In other words, our method outperforms others in maintaining robustness during pruning.

Table 4: Quantitative Analysis of Distance from Sentence Embeddings. We compare the distances between sentence embeddings derived from various layers of dense and sparse language models. Our findings reveal that our method aligns better with the dense model, regardless of whether we use the original or adversarial sentence. Refer to Figure 5 for a visualization of these sentence embeddings.

| Layer | Distance with dense | | | Data |
|---|---|---|---|---|
| | IMP + ADT (2x) | v.s. | Ours (2x) | |
| 1 | 0.0086 | > | **0.0000** | Ori |
| | 0.0086 | > | **0.0000** | Adv |
| 2 | 0.0144 | > | **0.0015** | Ori |
| | 0.0142 | > | **0.0015** | Adv |
| 3 | 0.0156 | > | **0.0014** | Ori |
| | 0.0258 | > | **0.0012** | Adv |
| 4 | 0.0193 | > | **0.0017** | Ori |
| | 0.0407 | > | **0.0017** | Adv |
| 5 | 0.0324 | > | **0.0067** | Ori |
| | 0.1319 | > | **0.0069** | Adv |
| 6 | 0.0763 | > | **0.0255** | Ori |
| | 0.0967 | > | **0.0253** | Adv |
| 7 | 0.1299 | > | **0.0496** | Ori |
| | 0.1478 | > | **0.0501** | Adv |
| 8 | 0.2530 | > | **0.1308** | Ori |
| | 0.2547 | > | **0.1078** | Adv |
| 9 | 0.1880 | > | **0.0958** | Ori |
| | 0.2767 | > | **0.0749** | Adv |
| 10 | 0.2804 | > | **0.1254** | Ori |
| | 0.3909 | > | **0.1049** | Adv |
| 11 | 0.4932 | > | **0.2322** | Ori |
| | 0.7317 | > | **0.0625** | Adv |
| 12 | 0.6872 | > | **0.2231** | Ori |
| | 0.6903 | > | **0.0349** | Adv |

### 5.6.2 Impact of Calibration Data

The calibration data is crucial for our methodology because it directly affects the computation of the Hessian Matrix. As outlined in Algorithm 1, the Hessian Matrix can be derived from $H = X^T X$. To further explore the impact of the number of data points, we designed experiments that gradually increased the number of data points used in our strategy. The results of these experiments are detailed in Figure 3. Our observations indicate that as the number of used data points increases, the robustness and accuracy of the sparse language modes increase, but only up to a certain threshold. We hypothesize that the model can initially retain

more general knowledge as data points increase. However, once a threshold is crossed where the new data cannot provide additional information for general features, adding more data points from a similar distribution no longer contributes to model robustness and accuracy.

### 5.6.3 Impact of Sparsity

As illustrated in Figure 4, we explore the robustness and accuracy of our sparse language models across a range of sparsity levels. In a departure from previous studies Zheng et al. (2022), our observations indicate that as sparsity increases, robustness decreases with a similar pace like accuracy. This trend suggests that the impact of increasing sparsity on model robustness might be less severe than previously assumed. This disparate pattern may stem from the post-training nature of our method. Furthermore, our observations regarding the trend in robustness align with the findings of previous studies by Zheng et al. (2022) and Liang et al. (2021). We note that the robustness of our sparse language models initially improves as sparsity escalates up to a certain threshold. After crossing this threshold, the robustness begins to decline. However, it sustains a level of robustness that is higher than the peak value observed in other models and does not collapse even with 10x compression. This finding further highlights the outstanding performance of our method in robust pruning.

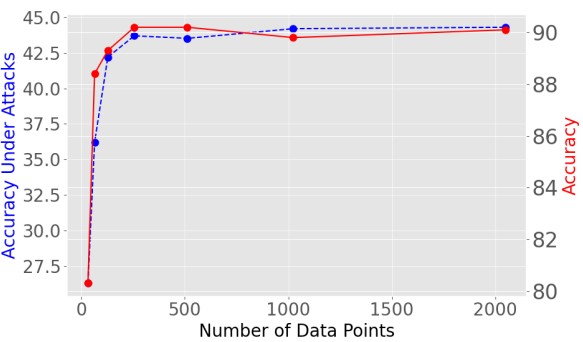

Figure 3: Impact of # of Calibration Data from SST2.

## 6 Conclusion

In this paper, we investigate the application of robust pruning methods for language models. We propose an adaptive pruning method and place a special emphasis on replicating the embedding and feature space of dense models to preserve as much pre-trained knowledge as possible. The effectiveness of this approach is confirmed through a series

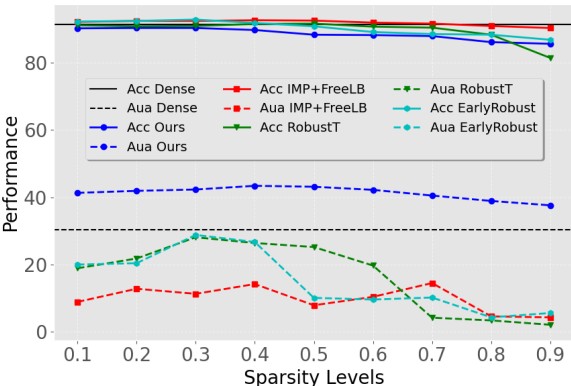

Figure 4: Impact of Sparsity Levels on SST2

of experiments conducted across various tasks.

## Limitations

This work introduces a post-training method that can robustly prune the language models without model retraining. Despite bypassing the rigorous retraining process, the computational cost of our method remains significant due to the calculation of the Hessian Matrix and its inverse. Consequently, this approach may not be feasible for language models comprised of billions of parameters. As a next step, we aim to refine our technique to devise a more efficient strategy to replicate the feature space and embedding space of language models

## Acknowledgements

The authors wish to thank the anonymous reviewers for their helpful comments.

## Ethics Statement

This work complies with the ACL Ethics Policy and we have carried out our research following the highest ethical standards. In our work on developing a new pruning strategy to enhance robustness in language models, we carefully considered the broader implications and ethical dimensions of this innovation.

While our research primarily concerns the improvement of model accuracy, sparsity, and robustness, we acknowledge that the use of these enhanced models can potentially be dual-use, which means they can be applied in both beneficial and harmful ways. An improved model can contribute positively by enhancing various NLP applications such as text summarization, machine translation, and sentiment analysis, potentially increasing efficiency and the overall quality of output. Fur-

thermore, these advancements could contribute to reducing the computational resources required for training and using large language models, which aligns with efforts to reduce the environmental impact of machine learning.

However, the increased robustness of models against adversarial attacks could also be used maliciously if the technology falls into the wrong hands. Bad actors could potentially exploit robust models for the generation of disinformation or manipulation of public sentiment, for instance. Furthermore, although our technique aims to faithfully replicate the feature space of dense models, bias present in the original training data could be preserved in the pruned models. Consequently, decisions made based on the output of these models could perpetuate these biases.

We encourage the use of our findings and methods for applications that promote the public good and contribute to human welfare. Further, we recommend that researchers and practitioners using this technique take into account potential biases in their training data and consider strategies for minimizing their impact. In the future, we hope to conduct more research on mitigating bias and other ethical issues associated with our pruning strategy. It is our belief that technology should be developed and used in a way that is transparent, fair, and beneficial to all.

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

## A Appendix-A

### A.1 Pruning with Hessian Matrix

As described in Section 3.2, we prune each layer of language models in a layer-wise setting. It involves an iterative step that removes a single weight for each step and updates the remaining weights until the desired sparsity level is attained. While this approach yields a locally optimal solution, it involves a computationally expensive step: calculating the Hessian matrix at each iteration. It is important to note that storing the information for a Hessian Matrix, denoted as $H$, requires $d \times d$ memory, and updating it has a computational complexity of $O(d^4)$, where $d = d_{row} \cdot d_{col}$.

### A.2 Accelerated Pruning with Hessian Matrix

Previous research highlights that the Hessian values across different rows of the weight matrix are independent. This is because the removal of a single weight in each row of the matrix only affects its corresponding row value. Consequently, we can simplify the objective function with $\sum_{i=1}^{d_{row}} \|W_{i,:}X - \hat{W}_{i,:}X\|_2^2$, and a separate Hessian Matrix of appropriate size ($d_{col} \times d_{col}$) for each row is sufficient to locate the optimal weight for removal. Additionally, since the output $Y = WX$ of the dense layer remains fixed, and the objective function for each row takes the standard form of least squares, its Hessian Matrix can be calculated by $H = 2XX^T$ (Frantar and Alistarh, 2022).

As the Hessian Matrix $H$ is no longer dependent on the weight, we only need to compute $H$ once. After each pruning step, the Hessian Matrix $H_M$ (M means the operation of removing or masking one single weight) can be obtained by masking the value at the corresponding location. However, when it comes to $H^{-1}$, the aforementioned trick cannot be applied as $(H^{-1})_M \neq (H_M)^{-1}$, making the computation still expensive. Frantar and Alistarh (2022) uses the Gaussian elimination technique for a more efficient update of $H^{-1}$. A mathematical exposition of this technique is provided below:

$$H_{-p}^{-1} = (H^{-1} - \frac{1}{[H^{-1}]_{pp}} H_{:,p}^{-1} H_{p,:}^{-1})_{-p} \quad (6)$$

where $-p$ meas remove single weight at index $p$. For more comprehensive details, please refer to the work of Frantar and Alistarh (2022).

## B Appendix-B

### B.1 Efficiency Analysis of Hessian Matrix

We recognize the importance of addressing the efficiency concern related to Hessian Matrix calculation. However, grasping the intricate balance between computational complexities and their broader implications is crucial. To provide clarity, we offer an in-depth analysis of computational complexities from both micro and macro viewpoints, contrasting it with approaches that necessitate model retraining.

### B.2 Micro Perspective

When considering models like $\text{Bert}_{base}$ and $\text{Bert}_{large}$, the computational requirements for the Hessian Matrix of one layer do not exceed that of model retraining in most cases. To clarify it, we analyze the complexity of our method step by step based on the Algorithm 2.

---

**Algorithm 2** Prune a linear layer $l$ of BERT with target sparsity $s$ and calibration data $X$

---

1: **Input:** Collect original $X, W, Y$ for $l$.
2: **procedure** PRUNING($l$)
3:     Set $W, X, Y \leftarrow l$
    **Adaptive Update 1:**
4:     Calculate $H^{-1} \leftarrow (XX^T)^{-1}$
5:     Set $W \leftarrow H^{-1}X^TY$
    **Pruning with Hessian Matrix:**
6:     Set $d_{in} \leftarrow$ input dimension.
7:     Set $k \leftarrow \text{int}(d_{in} \cdot s)$.
8:     **for** $j = 1$ **to** $k$ (parallel in rows) **do**
9:         Set $p \leftarrow argmin_{p \in d_{in}} \frac{1}{[H^{-1}]_{pp}} \cdot [W]_p^2$.
10:        Set $W \leftarrow W - [H]_{:,p}^{-1} \frac{1}{[H^{-1}]_{pp}} \cdot [W]p$.
11:        Set $A \leftarrow [H]_{:,p}^{-1}$
12:        Set $B \leftarrow [H]_{p,:}^{-1}$
13:        Set $H^{-1} \leftarrow H^{-1} - \frac{1}{[H^{-1}]_{pp}} AB$
14:        Remove $[W]_p$ from $W$
15:     **end for**
    **Adaptive Update 2:**
16:     Set $Y \leftarrow WX$.
17:     Update $X$ of next layer with post-process($Y$)
18: **end procedure**

---

**Notations:** To facilitate the understanding, we first introduce the notations essential for the complexity analysis. The sparsity ratio, a value lying between 0 and 1, is denoted by $s$. The input dimension of the linear layer is represented by $d_{in}$, and the output

| Methods | #Param | Re-T | SST2 | | | AGNEWS | | | IMDB | | |
|---|---|---|---|---|---|---|---|---|---|---|---|
| Fine-tune | 85M | Y | **92.3** | 12.7 | 86.2 | 94.7 | 19.1 | 80.0 | 95.1 | 7.4 | 92.2 |
| FreeLB | 85M | Y | 91.5 | 28.3 | 69.1 | **94.8** | 37.8 | 60.1 | 94.3 | 36.2 | 61.6 |
| Weight Average | 85M | Y | 91.4 | **30.4** | **66.75** | 94.4 | **48.5** | **48.6** | **95.2** | **44.4** | **53.4** |
| *sparsity = 50%* | | | | | | | | | | | |
| EarlyRobust (Stru) | 43M | Y | **91.2** | 15.6 | 82.9 | **94.1** | 28.4 | 69.8 | 90.7 | 33.2 | 63.3 |
| Ours (w/o Stru) | 43M | N | 88.31 | 43.1 | 51.2 | 93.4 | 48.5 | 48.1 | 94.2 | 53.2 | **43.6** |
| Ours (Stru 32:64) | 43M | N | 88.42 | **44.3** | 49.9 | 93.2 | **49.1** | 47.3 | **94.8** | **53.4** | 43.7 |

Table 5: Summary of Adversarial Robustness Assessment on BERT$_{base}$ in Structured Pruning. "Stru 32:64" refers to a pruning strategy where, for every 64 continuous weights (a bank) in a weight matrix, 32 of them are retained.

dimension, aligning with the weight matrix's other dimension, is symbolized by $d_{out}$. We use $d = d_{in} \times d_{out}$ to illustrate the comprehensive size of the weight matrix. The batch size and the sequence length are, respectively, given by $n$ and seq.

**Adaptive Update (1):** In this phase, the matrix multiplication $XX^T$ plays a pivotal role. Given the dimensions of $X$ as $n \times$ seq, $d_{in}$ and that of $X^T$ as $d_{in}$, seq $\times n$, the resulting matrix has a shape of $d_{in} \times d_{in}$. This multiplication alone possesses a complexity of $O(n \times \text{seq} \times d_{in}^2)$. Additionally, matrix inversion is another vital step with a complexity of $O(d_{in}^3)$. The computation of $H_i^{-1} X_i^T Y_i$ further contributes to the complexity, having a magnitude of $O(n \times \text{seq} \times d_{in} \times d_{out})$.

**Pruning with the Hessian Matrix:** In this context, the outer loop spans $d_{out}$ iterations. Within each row of $W$, an inner loop determined by $k = \text{int}(d_{in} \times s)$ is executed. This loop comprises various operations with $O(d_{in}^2)$. Summing up, the inner loop complexity is $O(k \times d_{in}^2)$. Consequently, the combined complexity for the pruning phase is $O(d_{in} \times s \times d_{in}^2 \times d_{out})$, simplifying to $O(d_{in}^3 \times s \times d_{out})$.

**Adaptive Update (2):** The matrix multiplication $Y = WX$ dominates with a complexity of $O(n \times \text{seq} \times d_{in} \times d_{out})$. Summing complexities for a single layer yields $O(2n \times \text{seq} \times d_{in} \times d_{out} + n \times \text{seq} \times d_{in}^2 + 2d_{in}^3 + d_{in}^3 \times s \times d_{out})$, with the dominant terms being $O(d_{in}^3 \times d_{out})$. Thus, pruning a layer has a complexity of $O(d_{in}^3 \times d_{out})$, which is also proved by Frantar and Alistarh (2022).

**Key observations:** A pivotal observation is that this complexity remains uninfluenced by the batch size $n$ because calibration data keeps $n$ restricted to a constant fall in $[128, 1024]$. The cubic relationship with $d_{in}$ is the primary driver behind the complexity, and for larger $d_{in}$, this can escalate

substantially.

## B.3 Comparison with Re-Training Method

In contrast, when training a single layer using SGD, the complexity is approximately $O(n \times \text{seq} \times d_{in} \times d_{out})$. This complexity scales linearly with the batch size $n$, which can increase markedly with large datasets and the number of training epochs. Although the complexity of the pruning operation remains consistent regardless of $n$, the training complexity escalates, posing computational challenges for extensive datasets, prolonged sequences, and increased training epochs. We also dive deeper into the comparative insights.

**Batch Size:** Our pruning method capitalizes on calibration data, thus constricting $n$ to moderate values, notably between 128 to 1024. This sharply diverges from the conventional training paradigm where $n$ can inflate significantly due to extensive datasets and number of training epochs, thereby magnifying its computational requisites.

**Dimensionality Dependency:** The intrinsic complexity of our pruning algorithm reveals a cubic dependency on $d_{in}$. This can render it computationally onerous, especially for layers endowed with an extensive $d_{in}$. Conversely, traditional training exhibits a linear correlation with both $d_{in}$ and $d_{out}$.

In summary, the computational demands of our pruning method, particularly for layers with a large $d_{in}$, are unquestionably stringent. However, it's important to recognize the significant computational burden introduced by traditional training, mainly because of its responsiveness to large dataset sizes. Understanding this balance and trade-off is crucial when comparing the effectiveness and suitability of our pruning approach to traditional retraining.

| Method | Dataset | Attack | Sparsity | Accuracy | Accuracy under attack |
|--------|---------|--------|----------|----------|----------------------|
| **Ours** | SST2 | TextBugger | 2x | 88.31% | **50.34%** |
| RobustT | SST2 | TextBugger | 2x | 90.5% | 35.6% |
| EarlyRobust | SST2 | TextBugger | 2x | 91.2% | 36.7% |
| **Ours** | SST2 | TextBugger | 4x | 86.93% | **49.08%** |
| **Ours** | SST2 | TextBugger | 8x | 85.6% | **48.85%** |
| **Ours** | SST2 | BERT-Attack | 2x | 88.31% | **51.95%** |
| RobustT | SST2 | BERT-Attack | 2x | 90.5% | 28.3% |
| EarlyRobust | SST2 | BERT-Attack | 2x | 91.2% | 30.2% |
| **Ours** | SST2 | BERT-Attack | 4x | 86.93% | **50.57%** |
| **Ours** | SST2 | BERT-Attack | 8x | 85.6% | **49.32%** |
| **Ours** | IMDB | TextBugger | 2x | 94.2% | **58.2%** |
| RobustT | IMDB | TextBugger | 2x | 93.2% | 46.1% |
| EarlyRobust | IMDB | TextBugger | 2x | 90.7% | 48.7% |
| **Ours** | IMDB | BERT-Attack | 2x | 94.2% | **52.1%** |
| RobustT | IMDB | BERT-Attack | 2x | 93.2% | 43.1% |
| EarlyRobust | IMDB | BERT-Attack | 2x | 90.7% | 43.5% |
| **Ours** | AGNews | TextBugger | 2x | 93.2% | **62.0%** |
| RobustT | AGNews | TextBugger | 2x | 94.8% | 44.1% |
| EarlyRobust | AGNews | TextBugger | 2x | 94.1% | 46.2% |
| **Ours** | AGNews | BERT-Attack | 2x | 93.2% | **70.8%** |
| RobustT | AGNews | BERT-Attack | 2x | 94.8% | 36.8% |
| EarlyRobust | AGNews | BERT-Attack | 2x | 94.1% | 39.3% |

Table 6: Evaluation of various methods and datasets against different adversarial attacks.

## B.4 Macro Perspective

**Predicable Processing Time and Promised Output**: Notably, from a broader view, while our approach introduces a dependency for each layer and potentially increases processing times, the number of layers in common language models is limited. This suggests that we can accurately predict the time needed to complete the pruning process, and expect positive results in return.

**Layer-by-Layer Computation for Resource Efficiency:** While the sum of Hessian Matrix computations of the entire language model is time-intensive, our approach uniquely addresses this by employing a layer-by-layer resolution strategy. This methodology means there's no necessity to simultaneously load the entire model into the memory of computational resources. Consequently, from a memory allocation standpoint, our pruning with the Hessian Matrix can be viewed as a resource-saving measure.

**Efficient Post-training Pruning:** A post-training pruning strategy is at the heart of our methodology. Unlike many other approaches that might require recurrent training sessions or exhaustive reiterations, ours stands out in its ability to save significant resources by strategically avoiding these processes.

**Computational Commitment:** While it's acknowledged that pruning with the Hessian Matrix does possess computational time costs, it's paramount to understand our larger vision. The ultimate objective isn't merely to save time but to sculpt a model characterized by three pillars: sparsity, robustness, and high performance. Such a model offers considerable advantages in real-world scenarios. Thus, the computational expenses encountered in the training phase can be viewed less as costs and more as strategic investments.

## C Appendix-C

### C.1 More Adversarial Attacks

To demonstrate the superiority of our method, we have incorporated further experiments targeting two more recognized adversarial attacks: BERT-Attack and TextBugger (Li et al., 2020b, 2018). BERT-Attack, powered by BERT, guarantees fluency and retains semantics in its adversarial outputs. Conversely, TextBugger integrates both character and word-level perturbations to yield adversarial instances, thereby introducing a new set of challenges for our defense mechanism. We use state-of-the-art methods (RobustT and EarlyRobust) as baselines and describe the results in Table 6 (Zheng

et al., 2022; Xi et al., 2022). Our approach consistently demonstrated superiority in the robustness of sparse language models across various sparsity levels and datasets.

## C.2 More Pruning Baseline

As recommended by the reviewer, we have included Movement Pruning (Sanh et al., 2020) as an additional baseline in our experiments. Our original selection of baselines was grounded on their capacity to simultaneously address accuracy, sparsity, robustness, and pruning cost. It should be noted that Movement Pruning predominantly emphasizes accuracy and sparsity.

Nevertheless, to offer a complete perspective, we have included Movement Pruning in our experimental evaluation. The comparative results are presented in Table 7. It is evident that, while our method may trail slightly in terms of clean accuracy, it significantly outperforms Movement Pruning under adversarial conditions, highlighting the robustness of our approach.

## D Appendix-D

### D.1 More Implementation Details

We utilize various hyperparameters and settings to fine-tune multiple downstream models for each dataset. The hyperparameters and settings employed are presented in Table 8. Subsequently, we apply the technique of *weight average* in a greedy manner to derive robust and dense models. The detailed procedure is outlined step-by-step in Algorithm 3.

---

**Algorithm 3** Greedy Weight Averaging

1: **procedure** GREEDYWA($\{h_1, \ldots, h_k\}$)
2: $\quad \{\theta_1, \ldots, \theta_k\} \leftarrow \{h_1, \ldots, h_k\}$
3: $\quad \{m_1, \ldots, m_k\} \leftarrow \{\theta_1, \ldots, \theta_k\}$
4: $\quad$ Sort($\{\theta_1, \ldots, \theta_k\}$) with $\{m_1, \ldots, m_k\} \downarrow$
5: $\quad ingredients \leftarrow \emptyset$
6: $\quad$ **for** $i = 1$ **to** $k$ **do**
7: $\qquad$ **if** Eval($average(ingredients \cup \{\theta_i\})$) $\geq$
8: Eval($average(ingredients)$) **then**
9: $\qquad\qquad ingredients \leftarrow ingredients \cup \{\theta_i\}$
10: $\qquad$ **end if**
11: $\quad$ **end for**
12: $\quad$ **return** $average(ingredients)$
13: **end procedure**

---

We adopt Textattack (Morris et al., 2020) to implement the method of adversarial attacks. Moreover, Aua% and Suc% are evaluated on all 872 test examples for SST-2, 500 randomly selected test samples for IMDB and AG NEWS.

The number of calibration data in our main experiments ranges from 256 to 1024 sentences. During pruning, we conduct our experiments on a server with a single NVIDIA 3090 GPU. Due to the layer-wise setting, we do not need to occupy substantial GPU memory, and our adaptive rule enables us to obtain an end-to-end rectification effect similar to SGD optimization.

### D.2 Impact of Structured Pruning

Drawing inspiration from the work by Xi et al. (2022), we also investigate the impact of structured pruning in our strategy. In particular, we evaluate our method's performance under N:M structured patterns and summarize the results in Table 4. We made several key observations from these experiments: 1) our method consistently produces better robust pruning results than other robust pruning methods in the context of structured pruning. 2) As proven by Xi et al. (2022), structured pruning enhances the robustness of subnetworks in comparison to unstructured pruning. Our experiments confirm the positive impact of structured patterns in pruning, solidifying the effectiveness of our robust pruning method.

## E Appendix-E

### E.1 Model Pruning

Pruning aims to eliminate redundant elements in neural networks, traditionally targeting elements of the smallest magnitude, which includes weights, output sensitivity, gradients, and Hessian matrices of training loss, among others. Pruning pre-trained language models like BERT has been an active field of research. Prasanna et al. (2020) demonstrated that unstructured pruning yields more sparse and accurate models. Pruning at the pre-training stage has been favored by researchers like Gordon et al. (2020) and Chen et al. (2021), due to its efficiency and effective knowledge transfer to downstream tasks. Sanh et al. (2020) adds penalty terms to the loss function to eliminate redundant weights. Frantar and Alistarh (2022) introduce an effective post-training pruning method, which is the first approach that prunes a language model in a one-shot manner without significant degradation in accuracy. However, these studies neglect robustness, focusing mainly on the accuracy-sparsity trade-off. Recent work has begun to note the issue of robustness for sparse language models, but the challenge of enhancing robustness with increased sparsity per-

Table 7: Comparison between our method and Movement Pruning under various attacks and sparsity levels.

| Method | Dataset | Attack | Sparsity | Accuracy | Accuracy under attack |
|---|---|---|---|---|---|
| Ours | SST2 | TextFooler | 2x | 88.31% | **43.12**% |
| Movement Pruning | SST2 | TextFooler | 2x | 90.6% | 14.85% |
| Ours | SST2 | TextFooler | 4x | 86.93% | **40.15**% |
| Movement Pruning | SST2 | TextFooler | 4x | 90.5% | 8.27% |
| Ours | SST2 | TextFooler | 8x | 85.6% | **37.63**% |
| Movement Pruning | SST2 | TextFooler | 8x | 90.0% | 9.14% |
| Ours | SST2 | TextBugger | 2x | 88.31% | **50.34**% |
| Movement Pruning | SST2 | TextBugger | 2x | 90.6% | 24.85% |
| Ours | SST2 | TextBugger | 4x | 86.93% | **49.08**% |
| Movement Pruning | SST2 | TextBugger | 4x | 90.5% | 21.35% |
| Ours | SST2 | TextBugger | 8x | 85.6% | **48.85**% |
| Movement Pruning | SST2 | TextBugger | 8x | 90.0% | 15.13% |

| ids | lr | opt | seed | epoc | wd | adt |
|---|---|---|---|---|---|---|
| #1 | 2e-5 | Adam | 42 | 10 | 1e-2 | Y |
| #2 | 3e-5 | AdamW | 426 | 20 | 1e-2 | N |
| #3 | 5e-5 | SGD | Random | 30 | 1e-2 | Y |
| #4 | 2e-5 | AdamW | 302 | 10 | 1e-3 | N |
| #5 | 4e-2 | AdamW | Random | 30 | 1e-2 | Y |
| #6 | 5e-5 | SGD | 42 | 3 | 1e-2 | N |
| #7 | 1e-5 | AdamW | 107 | 20 | 1e-3 | Y |
| #8 | 3e-5 | Adam | Random | 5 | 1e-2 | N |
| #9 | 2e-5 | AdamW | 302 | 30 | 1e-3 | Y |
| #10 | 2e-5 | SGD | Random | 15 | 1e-2 | N |

Table 8: A Range of Hyperparameters and Settings for Weight Averaging

sists (Zheng et al., 2022; Du et al., 2023; Xu et al., 2021; Liang et al., 2021; Xi et al., 2022), and the underlying causes of low robustness in language models remain elusive.

### E.2 Post-Training Pruning

Pruning methods can be categorized into Post-Training Pruning and In-Training Pruning according to if the pruning methods need extra model retraining. In the former, we are given a trained but uncompressed model, together with a small amount of calibration data. we must produce an accurate compressed model in one shot, i.e., a single compression step, without retraining and with limited computational costs. This is motivated by practical scenarios such as the large language models, which are hard to train or even finetune because of the complicated training process. In this paper, our method is a Post-Training pruning method.

### E.3 Layer-wise Pruning

Layerwise Pruning is an important approach to optimizing language models, offering a distinct methodology compared to end-to-end pruning. Unlike end-to-end pruning, which simultaneously evaluates and prunes the entire model as a whole, layerwise pruning tackles each layer of the neural network individually. This means pruning decisions are based on a layer-specific analysis, often using a metric like the magnitude of the weights to determine which parameters within that layer are least significant and can be removed without substantially impacting the layer's output. By selectively reducing the number of parameters in each layer, layerwise pruning can effectively decrease the computational requirements and memory footprint of language models while maintaining their accuracy. The layerwise approach offers an advantage in that it provides a more granular level of control over the pruning process, which can be beneficial in preserving model performance while achieving efficiency gains.

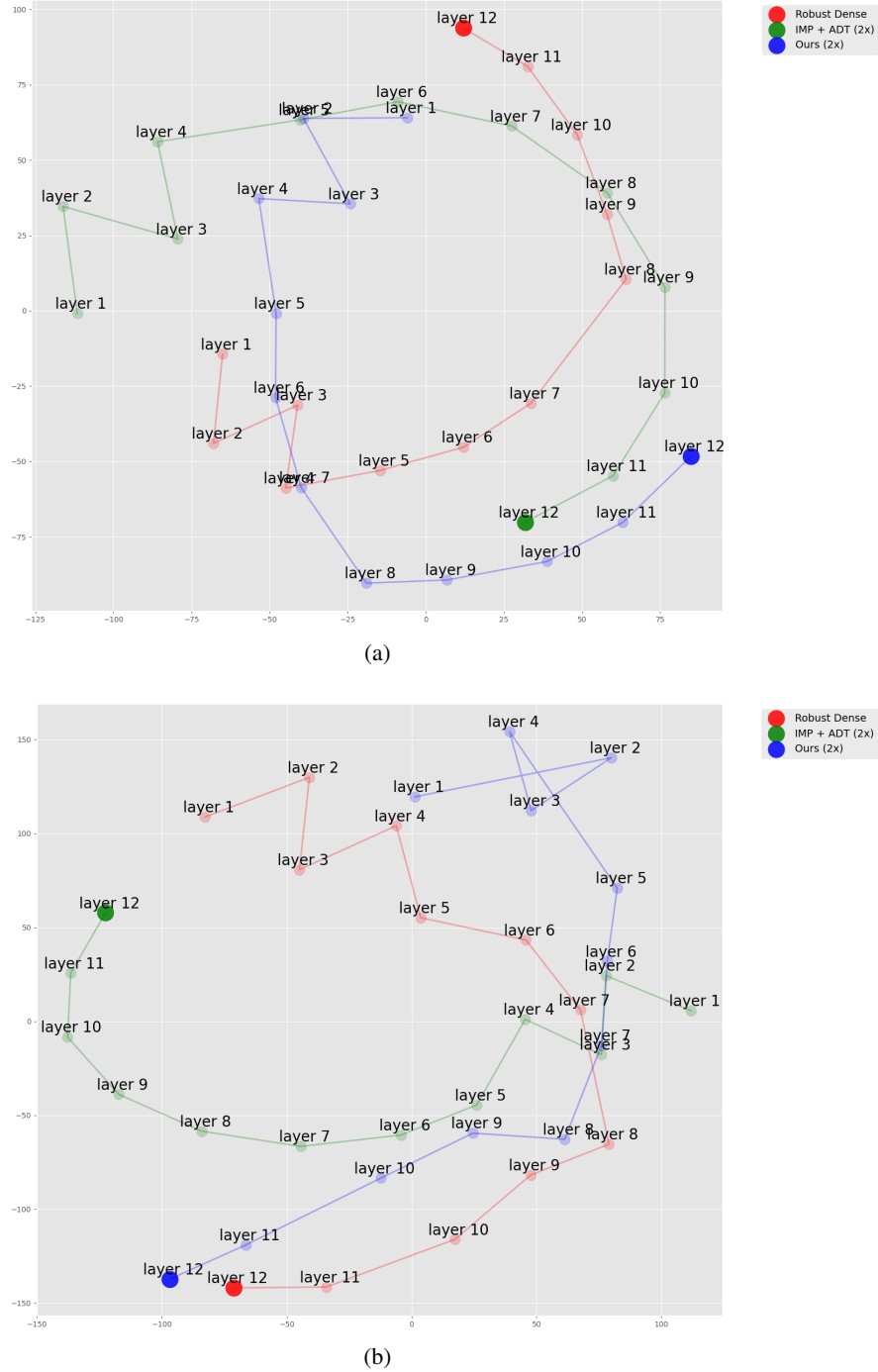

Figure 5: Visualization of Sentence Embeddings. We've compared the distance of sentence embeddings between the robust and dense model (red), the sparse language models generated with IMP+FreeLB (green), and the sparse language models created using our method (blue). Figure 5a displays the two-dimensional representation of the embeddings from different layers of various models for sentence i ("*allows us to hope that nolan is prepped to embark on a major career as a commercial yet shrewd scriptwriter*"). Similarly, Figure 5b showcases the two-dimensional representation of the embeddings from different layers of various models for sentence ii ("*allows us to hope that nolan is poised to embark on a major career as a commercial yet inventive filmmaker*"). Note that sentence i originates from SST2 dataset, and all three models accurately predict its label. On the other hand, sentence ii, an adversarial sample generated from sentence i, is only correctly predicted by the robust and dense model and our sparse language model. We use the embedding of the first token ([CLS]) as the representation of sentences, as the model uses this for the final classification. **Clearly, our method can generate embeddings and features that align more closely with the robust and dense model under adversarial attacks.**