# OpenReview forum: "Towards Robust Pruning: An Adaptive Knowledge-Retention Pruning Strategy for Language Models"
_EMNLP/2023/Conference — EMNLP 2023 Main_

### Official Review · Reviewer_1hiE · 2023-07-28

**Typos Grammar Style And Presentation Improvements:** The limitations of exiting methods sh…
**Soundness:** 3

**Excitement:**

3: Ambivalent: It has merits (e.g., it reports state-of-the-art results, the idea is nice), but there are key weaknesses (e.g., it describes incremental work), and it can significantly benefit from another round of revision. However, I won't object to accepting it if my co-reviewers champion it.

**Paper Topic And Main Contributions:**

This paper introduces a pruning strategy, which can have a balance between accuracy, sparsity, robustness, and pruning cost. More specifically, they firstly use weight averaging to obtain a robust dense model. Then, an adaptive pruning method is adopted to obtain sparse models. The experimental results demonstrate the effectiveness of their proposed method.

**Questions For The Authors:**

1. Could you conclude the limitations in existing methods? Why is the proposed model able to outperform existing models?

2. One of the proposed techniques, weight averaging, requires obtaining several different versions of a model. Will it increase the computation cost?

**Reasons To Accept:**

1. The research topic is important and meaningful.

2. According to the experimental results, the proposed method is effective.


**Reasons To Reject:**

1. The limitations in existing methods are not introduced clearly. It is not easy for people who are not working on pruning to understand the paper.

2. The method needs to obtain several different models for weight averaging. It will increase the computation cost.


**Reproducibility:**

3: Could reproduce the results with some difficulty. The settings of parameters are underspecified or subjectively determined; the training/evaluation data are not widely available.

**Reviewer Confidence:**

2: Willing to defend my evaluation, but it is fairly likely that I missed some details, didn't understand some central points, or can't be sure about the novelty of the work.

---

> ### Author Rebuttal · Authors · 2023-08-27
>
> We sincerely appreciate the reviewer's time and effort in evaluating our paper. Your comments and questions provide valuable insights that will aid in enhancing the quality and clarity of our work. Specifically, you've raised the following questions:
>
> - **Question 1**: Could you conclude the limitations in existing methods? Why is the proposed model able to outperform existing models?
> - **Question 2**: One of the proposed techniques, weight averaging, requires obtaining several different versions of a model. Will it increase the computation cost?
>
> Thank you for the review and for bringing up these questions again. Please see our responses below to clarify the main concerns. Also, if you feel that your original concerns have been resolved, we would appreciate it if you would update your evaluation to reflect this. Thank you!
>
> ## Question 1 （Limitation Related)
> ---
>
> Thank you for bringing up the clarifications regarding the limitations of existing methods and the distinctiveness of our proposed model.
>
> **Limitations in Existing Methods**:
>
> 1. **`[Line 058-063, 068-072]` Robustness-Accuracy Trade-off**: Many current methods, including those by \citet{zheng-etal-2022-robust} and \citet{xi-etal-2022-efficient}, struggle to maintain a balanced relationship between robustness (accuracy under adversarial conditions) and standard accuracy. Although some results on particular datasets may suggest optimal performance, a discernible vulnerability emerges in the disparity between standard and adversarially tested accuracy metrics.
>
> 2. **`[Line 068-072]` Limited Sparsity**: It's pertinent to mention that numerous prevailing robust pruning techniques achieve only modest sparsity, often less than a 2x compression rate. As sparsity exceeds 2x, these methods often witness a stark reduction in robustness and can even compromise clean accuracy.
>
> 3. **`[Line 234-237, 262-267]` Retraining Overheads**: A predominant limitation in many existing pruning strategies is the unavoidable retraining phase, which increases computational demands and extends the time to deployment for pruned models.
>
> **Why Our Method Outperforms Existing Works**
>
> 1. **Preservation of Pre-trained Knowledge**: We are anchored in the conviction that the robustness of a pruned model correlates directly with the extent of pre-trained knowledge it retains. Our methodology emphasizes replicating the embedding and feature spaces of the original full models, ensuring that pruned versions maintain substantial semantic insights. This meticulous preservation inherently enhances the model's robustness, facilitating adept handling of sophisticated semantic intricacies even under adversarial onslaughts.
>
> 2. **Strategic Layer-by-Layer Pruning**: Our pruning methodology is sequential and tailored to individual layers. This granular strategy minimizes the forfeiture of critical semantic features, ensuring the pruned model's functional integrity.
>
> 3. **Adaptive Error Rectification**: Recognizing that pruning one layer might inadvertently cascade errors, potentially amplifying pre-existing ones, our approach incorporates an adaptive error correction mechanism. This recursive refinement process ensures that the resulting pruned model harmoniously melds sparsity, robustness, and accuracy — and remarkably, sidesteps the need for exhaustive retraining.
>
> 4. **Post-training Pruning**: A hallmark of our model is its capability to prune post-training. This approach obviates the necessity for computationally intensive retraining, thereby economizing computational resources.
>
> In light of these innovations, our empirical evaluations persistently underscore our method's superiority over conventional benchmarks, predominantly in accuracy, sparsity, robustness, and pruning efficiency domains. By proactively addressing the intrinsic challenges of extant pruning strategies, our method paves the way for a more holistic, adept, and efficient paradigm in the realm of robust pruning for language models.
>
> ## Question 2 (Computation Cost of Weight Averaging):
>
> Thank you for your insightful inquiry regarding the computation cost introduced by our weight averaging technique. Let's delve deeper into this methodology's intricacies and address the associated computational implications:
>
> 1. **Acknowledging Increased Computation**:
>    Initially, we acknowledge that the weight averaging technique incurs an additional computational overhead. This method involves training multiple models and then averaging them to achieve a robust and dense model. This approach is designed to ensure a precise analytical foundation. By doing so, we can eliminate factors that may distort our experimental results interpretation.
>
> 2. **Purpose of Weight Averaging**:
>    Despite language models exhibiting commendable performance on specific datasets, they frequently rely on surface-level or spurious features rather than delving deep into sophisticated semantic nuances. This presents a dilemma when determining if the vulnerability of a pruned model stems from the inherent challenges of the dense model or the pruning techniques adopted. Our weight averaging strategy proactively addresses this issue by building a robust and dense foundation before any pruning.
>
> 3. **Inspiration & Approach**:
>    Our technique, inspired by \citet{croce2023seasoning} and \citet{wortsman2022model}, harnesses the power of weight averaging to forge robust language models. The core principle is to train multiple models with varied hyperparameters or settings. This variety enables each model to understand and capture unique data nuances. By strategically averaging their weights, we amalgamate this dispersed knowledge, leading to a stronger model.
>
> 4. **Greedy Weight Averaging**:
>    Conventional adversarial training tends to show unpredictable performance on language models. Our experiments have revealed unsatisfactory outcomes with these methods. In contrast, our greedy weight averaging approach consistently produces a stable, robust, and dense model. This strategy meticulously arranges models based on their performance under adversarial attacks and averages the weights that bolster the overall robustness, as elucidated in Algorithm~\ref{algo:2}.
>
> 5. **Efficient Pruning Strategy**:
>    Once we achieve a robust and dense language model, our main pruning strategy functions as a post-training method. This approach strategically eliminates the necessity for extensive training phases, thus efficiently conserving computational resources.
>
> 6. **Eyes on the Prize**:
>    The endgame is to craft a model that's sparse, robust, and primed for high performance, making it ideal for real-world deployments. Given this context, the clear benefits such a model presents in operational environments undoubtedly justify the computational expenses borne during the training phase. This up-front computational commitment is a strategic investment, poised to yield substantial returns in real-world applications.
>
> To sum it up, the computational increase due to weight averaging is a deliberate and tactical move, aimed at birthing a model that seamlessly merges robustness with practical efficiency.
>
> We will address these concerns comprehensively in the following sections.

---

### Official Review · Reviewer_YNs7 · 2023-08-08

**Soundness:** 3

**Excitement:**

4: Strong: This paper deepens the understanding of some phenomenon or lowers the barriers to an existing research direction.

**Missing References:**

Sanh, Victor, Thomas Wolf, and Alexander Rush. "Movement pruning: Adaptive sparsity by fine-tuning." Advances in Neural Information Processing Systems 33 (2020): 20378-20389.

Jain, Sarthak, and Byron C. Wallace. "Attention is not explanation." arXiv preprint arXiv:1902.10186 (2019).

**Paper Topic And Main Contributions:**

In this paper, the authors propose an adaptive tuning based on optimal brain damage, which utilizes the Hessian matrix to guide the pruning process. The proposed method looks interesting because the Hessian matrix can well characterize the curvature of the loss surface to best preserve the performance of pruned models, but it becomes obsolete given the parameter scale of modern neural networks. It is good to see the authors bring this method back based on some recent advances in efficiently estimating the Hessian matrix. However, I have some questions regarding the main content: After the pruning, does your method still need an additional tuning step to recover the performance loss? If so, I believe the authors should choose a more competitive baseline like Movement Pruning (Sanh et al., 2020), which retains only 3% parameters with minimal accuracy loss. The authors also claim the performance loss of pruning originates from the loss of pretrained knowledge. In Section 5.6.1, the authors use attention scores as a proxy of pretrained knowledge. Given the observations from recent research (Jain and Wallace, 2019), it may not be a reasonable choice and the authors should give a clearer definition. Last but not least, the writing of this paper could be improved, e.g., argmin in Eqs. (3) and (4) should not be in italics and ";" should be replaced by "s.t.".

**Questions For The Authors:**

1. Does your method need additional tuning to recover the performance loss after pruning?

**Reasons To Accept:**

1. The authors propose an interesting Hessian-matrix-based pruning method.
2. The experiments show that the proposed method preserves performance well and is robust to adversarial attacks.

**Reasons To Reject:**

1. The writing could be improved and some descriptions are not clear.
2. The chosen baseline seems not competitive.

**Reproducibility:**

3: Could reproduce the results with some difficulty. The settings of parameters are underspecified or subjectively determined; the training/evaluation data are not widely available.

**Reviewer Confidence:**

2: Willing to defend my evaluation, but it is fairly likely that I missed some details, didn't understand some central points, or can't be sure about the novelty of the work.

**Typos Grammar Style And Presentation Improvements:**

argmin in Eqs. (3) and (4) should not be in italics and ";" should be replaced by "s.t.".

---

> ### Author Rebuttal · Authors · 2023-08-26
>
> We sincerely appreciate the insightful comments and questions provided by the reviewer. Your feedback is invaluable for enhancing the quality and clarity of our work. To ensure we address each concern thoroughly, we've summarized the primary questions raised:
>
> 1. **Methodology and Additional Tuning**: Does our Hessian-matrix-based method require a post-pruning tuning step to recover performance?
> 2. **Choice of Baseline**: Why wasn't the Movement Pruning approach, which retains just 3% of parameters with minimal accuracy loss, chosen as a baseline?
> 3. **Origins of Performance Loss and Its Correlation to Pruning**: Could we further elucidate our claim regarding the performance loss post-pruning being attributed to the loss of pretrained knowledge? In light of recent research, notably, Jain and Wallace (2019), is using attention scores to represent pretrained knowledge justified?
> 5. **Writing and Stylistic Concerns**: Are there plans to address the stylistic issues, especially in Eqs. (3) and (4), where 'argmin' is in italics and ";" should be replaced with "s.t."?
>
> Thank you for the review and for bringing up these questions again. Please see our responses below to clarify the main concerns. Also, if you feel that your original concerns have been resolved, we would appreciate it if you would update your evaluation to reflect this. Thank you!
>
>  ## Question 1 (Methodology and Additional Tuning)
> ---
> We appreciate the reviewer's inquiry about post-pruning tuning. First and foremost, it is essential to emphasize that our method unequivocally does not require additional tuning after pruning to recover performance. This is a testament to our pruning approach's efficacy and stability.
>
> Beyond this point, our experiments did reveal an intriguing observation. Upon obtaining the pruning results, we found that minor fine-tuning could sometimes enhance outcomes regarding clean accuracy and accuracy under adversarial attacks. Yet, this improvement isn't universally consistent. However, prior work `(Xu et al., 2021, https://aclanthology.org/2021.emnlp-main.832.pdf)` claims that such fine-tuning could potentially compromise robustness. While our approach remains solid without needing post-pruning adjustments, we value the reviewer's insight, as it provides thoughtful considerations for future explorations.
>
> ## Question 2 (Choice of Baseline):
> ---
>
> We acknowledge the reviewer's suggestion of Movement Pruning (Sanh et al., 2020) as a potential competitive baseline. We initially did not prioritize this method because it primarily emphasizes accuracy and sparsity. Our selection of baselines was driven by works that holistically address accuracy, sparsity, robustness, and pruning cost simultaneously.
>
> However, to address the reviewer's concern and provide a comprehensive perspective, we revisited our experiments and included Movement Pruning in our evaluations. The results are showcased in the subsequent table. While our method might slightly lag in clean accuracy compared to this baseline, it's noteworthy that we outperform it in terms of accuracy under attack. This lead in robustness is significant and underscores the value and distinctiveness of our approach.
>
> | Method | Dataset   | Attack | Sparsity | Accracy | Accracy under attack |
> |----------|-----------|---------------|:----------------:|:----------:|:------------:|
> | Ours | SST2      | TextFooler   | 2x      |   88.31%   |  **43.12%**   |
> | Ours | SST2      | TextFooler   | 4x       |  86.93% |  **40.15%**  |
> | Ours | SST2      | TextFooler   | 8x       | 85.6% |  **37.63%**   |
> | Movement Pruning | SST2      | TextFooler   | 2x       | 90.6% |  14.85%   |
> | Movement Pruning | SST2      | TextFooler   | 4x       | 90.5% |  8.27%   |
> | Movement Pruning | SST2      | TextFooler   | 8x       | 90.0% |  9.14%   |
> | Ours | SST2      | TextBugger   | 2x      |   88.31%   |  **50.34%**   |
> | Ours | SST2      | TextBugger   | 4x       |  86.93% |  **49.08%**  |
> | Ours | SST2      | TextBugger   | 8x       | 85.6% |  **48.85%**   |
> | Movement Pruning | SST2      | TextBugger   | 2x       | 90.6% |  24.85%   |
> | Movement Pruning | SST2      | TextBugger   | 4x       | 90.5% |  21.35%   |
> | Movement Pruning | SST2      | TextBugger   | 8x       | 90.0% |  15.13%   |
>
> ## Question 3 (Origins of Performance Loss and Its Correlation to Pruning)
> ---
>
> Thank you for pinpointing a critical aspect of our work and challenging our choice of using attention scores to indicate pre-trained knowledge. Based on your concern and the cited work of Jain and Wallace (2019), we understand the importance of a more definitive measure. Here’s our revised approach:
>
> 1. **Acknowledgment of the Concern**:
>    - We acknowledge that solely relying on attention scores may not sufficiently represent the richness and breadth of pre-trained knowledge, and we appreciate the reference to the findings of Jain and Wallace (2019), which challenges its use as a singular metric.
>
> 2. **Delineation of Pre-trained Knowledge**:
>    - While we initially proposed attention scores as one metric, we understand that a more comprehensive definition is essential. Pre-trained knowledge encompasses not just attention patterns but semantic understanding, feature embeddings, and the model's ability to generalize across various inputs.
>
> 3. **Updated Experimental Approach**:
>    - **Experiment A - Feature Embedding Analysis**: We compare the embeddings produced by pruned and dense models for various inputs. The similarity in embeddings will indicate the retention of pre-trained knowledge. Specifically, we undertook a quantitative analysis to gauge the distance between the embeddings produced by our approach and those from the dense model. Across the board, our technique outperformed other pruning methods regarding feature distance. This distance measurement is detailed in Equation 3.
>
>        **Result Table:**
>
>        **`Distance for datapoint from original SST2 dataset`**
>        | Layer | Distance with IMP + ADT (2x) | Distance with Ours (2x) |
>        |-------|-------------------------|--------------------|
>        | 1 | 0.0086 | 0.0000 |
>        | 2 | 0.0144 | 0.0015 |
>        | 3 | 0.0156 | 0.0014 |
>        | 4 | 0.0193 | 0.0017 |
>        | 5 | 0.0324 | 0.0067 |
>        | 6 | 0.0763 | 0.0255 |
>        | 7 | 0.1299 | 0.0496 |
>        | 8 | 0.2530 | 0.1308 |
>        | 9 | 0.1880 | 0.0958 |
>        | 10 | 0.2804 | 0.1254 |
>        | 11 | 0.4932 | 0.2322 |
>        | 12 | 0.6872 | 0.2231 |
>
>        **`Distance for adversarial sample generated from above datapoint`**
>        | Layer | Distance with IMP + ADT (2x) | Distance with Ours (2x) |
>        |-------|-------------------------|--------------------|
>        | 1 | 0.0086 | 0.0000 |
>        | 2 | 0.0142 | 0.0015 |
>        | 3 | 0.0258 | 0.0012 |
>        | 4 | 0.0407 | 0.0017 |
>        | 5 | 0.1319 | 0.0069 |
>        | 6 | 0.0967 | 0.0253 |
>        | 7 | 0.1478 | 0.0501 |
>        | 8 | 0.2547 | 0.1078 |
>        | 9 | 0.2767 | 0.0749 |
>        | 10 | 0.3909 | 0.1049 |
>        | 11 | 0.7317 | 0.0625 |
>        | 12 | 0.6903 | 0.0349 |
>
>        **Our methodology consistently showcases superior performance, achieving minimized distance scores throughout all layers for both the original SST2 data points and the adversarial instances. This superiority is even more evident in scenarios involving adversarial attacks**, where our technique's embeddings are distinctly closer to the dense model in comparison to the IMP + ADT method.
>
>    - **Experiment B - Adversarial Robustness Comparison**: By challenging pruned models against adversarial attacks, we can better understand their semantic retention and robustness. This is certified by our main experiment,  and the pivotal results are presented in Table 1 in our paper.
>
> 4. **Clarification & Way Forward**:
>    - While our earlier approach provided a preliminary view, we aim to refine and expand our methodologies to validate our hypothesis more accurately. In addition to the aforementioned experiments, we are also open to other ways to holistically assess pre-trained knowledge's preservation.
>
> In summary, we sincerely appreciate the feedback. We are committed to refining our research methodologies to offer a clearer and more comprehensive insight into the effects of pruning on the retention of pre-trained knowledge in language models.
>
> ## Question 4 (Writing and Stylistic Concerns)
> ---
> We sincerely appreciate the reviewer's feedback on our paper's writing and presentation aspects. We acknowledge the points raised, including the improper italicization of "argmin" in Eqs. (3) and (4) and the suggested replacement of ";" with "s.t.". We are committed to enhancing the clarity and precision of our paper, and we will incorporate these suggested changes in our revision to ensure the manuscript meets high presentation standards.
>
> We are committed to comprehensively addressing these questions in our revised manuscript.

---

### Official Review · Reviewer_VnKg · 2023-08-09

**Soundness:** 3

**Excitement:**

3: Ambivalent: It has merits (e.g., it reports state-of-the-art results, the idea is nice), but there are key weaknesses (e.g., it describes incremental work), and it can significantly benefit from another round of revision. However, I won't object to accepting it if my co-reviewers champion it.

**Paper Topic And Main Contributions:**

The authors propose a new robust pruning method for LLMs that also is meant to improve the model's ability  to defend itself against adversarial attack.  They authors report improve results on adversarial robustness assessments vs. multiple existing models.

**Questions For The Authors:**

How does Figure 5 show that your method generates embeddings that align more closely?  Do you have quantitative results?

**Reasons To Accept:**

The results are strong against competitors, particularly on accuracy under attack and minimizing the success ratio of adversarial tasks. The method does not require retraining (though they admit in Limitations that their method is also computationally expensive because of "calculation of the Hessian and its inverse".  They compare to multiple models from the literature, and have improved performance.  They assess their method at multiple levels of sparsity.

**Reasons To Reject:**

The motivation of the paper seems to be to keep weights after pruning similar to in the dense model, however I do not see quantitative results on how similar the resulting model is. Seems like you could calculate this based on equation 4, and I'd like to see that result. Figure 2 of attention score visualization and Figure 5 of a sentence embedding are used as evidence, but these are only on 1-2 examples.   Figure 5 in particular says "Clearly, our method can generate embeddings and features that align more closely with the robust and dense model under adversarial attacks", but I do not see how one can conclude this from the image.

I think a bit of a reframing of the intro would be helpful.  The intro states "this paper investigates why language models are susceptible to adversarial attacks".  I wouldn't say that the paper addresses this question - rather it proposes one robust pruning method.  Question 1, "What is the core to defend against adversarial attacks for sparse language models" also goes unanswered. This issues can be corrected by reframing the paper.


**Reproducibility:**

3: Could reproduce the results with some difficulty. The settings of parameters are underspecified or subjectively determined; the training/evaluation data are not widely available.

**Reviewer Confidence:**

2: Willing to defend my evaluation, but it is fairly likely that I missed some details, didn't understand some central points, or can't be sure about the novelty of the work.

**Typos Grammar Style And Presentation Improvements:**

Figure 5 needs axis labels.

I recommend adding a few adversarial examples early on in the paper so the reader has a nice example of the task you're approaching.

I recommend a little more clarification on the math.  For instance, remind the reader what the Hessian is, and why we're using it.

In Equation 3, what is K?

Can you make Algorithm 1 - particularly the "Pruning with Hessian Matrix" section, more intuitive?

You mentioned that your method also requires a lot of computation.  How much, compared to retraining?

Figure 5 is mentioned in the main body of the paper, but is in fact in the appendix so should be relabeled, or moved.

Sectoin 5.6.2 "Impact of Calibration Data", could use more explanation. How are these data points used?

---

> ### Author Rebuttal · Authors · 2023-08-27
>
> We sincerely thank the reviewer for their insightful comments and questions, which have offered us a clear perspective on areas for improvement. We have carefully gone through the feedback and summarized the major points raised. Below, we have listed the primary concerns and questions from the review to address them systematically in our rebuttal.
>
> ---
>
> **Questions for the Authors:**
> - How does Figure 5 illustrate that your method generates embeddings that align more closely with the robust and dense model? Do you have any quantitative results to support this claim? (**We quantify the distance with the Equation 3**)
> - What does K represent in Equation 3?
> - Compared to retraining, how computationally intensive is your method?
> - Could you clarify the use of data points in the "Impact of Calibration Data" section?
>
> **Feedback and Suggestions:**
> - Reframing the introduction to reflect the paper's content more accurately is recommended.
> - Including a few adversarial examples at the beginning of the paper would provide a clearer context for readers.
> - Some sections, like those discussing the Hessian and Algorithm 1, would benefit from further clarification.
> - The placement and reference to Figure 5 need to be consistent.
>
> Thank you for the review and for bringing up these questions again. Please see our responses below to clarify the main concerns. Also, if you feel that your original concerns have been resolved, we would appreciate it if you would update your evaluation to reflect this. Thank you!
>
> ## Questions for the Authors
> ---
>
> ### Question 1:
> How does Figure 5 illustrate that your method generates embeddings that align more closely with the robust and dense model? Do you have any quantitative results to support this claim? (**We quantify the distance with the Equation 3**)
>
> ---
>
> **Acknowledgment:** We understand the confusion stemming from the qualitative nature of Figure 5.
>
> **Resolution:** In response to this concern, we undertook a quantitative analysis to gauge the distance between the embeddings produced by our approach and those from the dense model. Across the board, our technique outperformed other pruning methods regarding similarity scores. This distance measurement is detailed in Equation 3.
>
> **Result Table:**
>
> **`Distance for datapoint from original SST2 dataset`**
> | Layer | Distance with IMP + ADT (2x) | Distance with Ours (2x) |
> |-------|-------------------------|--------------------|
> | 1 | 0.0086 | 0.0000 |
> | 2 | 0.0144 | 0.0015 |
> | 3 | 0.0156 | 0.0014 |
> | 4 | 0.0193 | 0.0017 |
> | 5 | 0.0324 | 0.0067 |
> | 6 | 0.0763 | 0.0255 |
> | 7 | 0.1299 | 0.0496 |
> | 8 | 0.2530 | 0.1308 |
> | 9 | 0.1880 | 0.0958 |
> | 10 | 0.2804 | 0.1254 |
> | 11 | 0.4932 | 0.2322 |
> | 12 | 0.6872 | 0.2231 |
>
> **`Distance for adversarial sample generated from above datapoint`**
> | Layer | Distance with IMP + ADT (2x) | Distance with Ours (2x) |
> |-------|-------------------------|--------------------|
> | 1 | 0.0086 | 0.0000 |
> | 2 | 0.0142 | 0.0015 |
> | 3 | 0.0258 | 0.0012 |
> | 4 | 0.0407 | 0.0017 |
> | 5 | 0.1319 | 0.0069 |
> | 6 | 0.0967 | 0.0253 |
> | 7 | 0.1478 | 0.0501 |
> | 8 | 0.2547 | 0.1078 |
> | 9 | 0.2767 | 0.0749 |
> | 10 | 0.3909 | 0.1049 |
> | 11 | 0.7317 | 0.0625 |
> | 12 | 0.6903 | 0.0349 |
>
> **Results indicate that our method consistently achieves lower distance scores across all layers for original SST2 data points and adversarial samples. This phenomenon becomes even more pronounced under adversarial attack scenarios.** Specifically, our technique's embeddings are closer to the dense model than those generated using the IMP + ADT  method.
>
> ## Question 2:
> What does K represent in Equation 3?
>
> ---
>
> Thank you for pointing out the ambiguity regarding the representation of \(K\) in Equations 3 and 4. We sincerely apologize for the oversight.
>
> To address your question: in the context of the paper, the goal is to find sparse parameters \(\hat{W}_l\) that align closely with the dense model while ensuring robustness against adversarial attacks. This alignment is quantified by minimizing the discrepancy between the dense and sparse layer outputs. The term \(k\) in the provided equations sets a constraint on the sparsity of the pruned model. Specifically, \(k\) dictates the maximum number of non-zero weights (or parameters) the pruned model is allowed to retain.
>
> **In simple terms, \(K\) in Equations 3 and 4 designates the total number of weights that remain non-zero after the pruning process, thus serving as a threshold to control the model's sparsity.**
>
> We realize the importance of clarifying this in our paper, and we will ensure that this point is revised and made more explicit in the subsequent version. Again, we appreciate your valuable feedback, and we apologize for any confusion caused.
>
> ### Question 3:
> Compared to retraining, how computationally intensive is your method?
>
> ---
>
> Our approach encompasses two main components: the **Average Weighting for Robust Dense Model** and the **Adaptive Pruning for Robust Sparse Model**. For clarity and depth, we will individually discuss each component in comparison to the retraining method. Specifically, our examination will be bifurcated into time (macro-level) and computational workload (micro-level).
>
> 1. **Average Weighting for Robust Dense Model**:
>
>     - **Purpose and Flexibility**: The weight averaging process for achieving a robust dense model is not a mandatory component for our robust pruning approach. Any effective and swift method to generate a robust dense model is compatible with our core methodology. The choice of average weighting has been made based on its relative stability compared to the more erratic behavior of traditional adversarial training in the realm of NLP. In essence, average weights offer a greedy yet stable means to consistently produce a robust dense model.
>
>     - **Macro Perspective - Time**: The models used for weight averaging operate independently, without inter-dependencies. As such, there's no cumulative time lag due to the sequencing of model execution.
>
>     - **Micro Perspective - Computation**: The computational demand for average weights primarily depends on the number of hyperparameter sets. Notwithstanding, it’s pivotal to understand that while average weights provide stable robust dense models, they can be readily substituted by any other efficient method that swiftly produces a robust dense model.
>
> 2. **Adaptive Pruning for Robust Sparse Model**:
>
>     - **`Micro Perspective - Computation`**: When considering models like Bert_base and Bert_large, the computational requirements for the Hessian Matrix of one layer do not exceed that of model retraining in most cases. To clarify it, we analyze the complexity of our method step by step based on one linear layer.
>
>       **Algorithm**: Prune a linear layer $l$ of BERT with target sparsity `s` and calibration data `X`
>
>       1. **Input:**
>           - Collect original $X$, $W$, $Y$ for $l$.
>
>       2. **Procedure:** Pruning $l$
>
>           1. Set $W$, $X$, $Y$ ← $l$
>
>           ---
>
>           - _Adaptive Update_:
>
>               1. Calculate $H^{-1}$ ← $(XX^{T})^{-1}$
>               2. Set $W$ ← $H^{-1}X^{T}Y$
>
>           ---
>
>           - _Pruning with Hessian Matrix_:
>
>               1. Set $d_{in}$ ← input dimension.
>               2. Set $k$ ← int($d_{in} \cdot s$).
>               3. **for** $j$ from 1 to $k$ (parallel in rows):
>                   1. Calculate $p$ ← $argmin_{p\in{d_{in}}}\frac{1}{[H^{-1}]{pp}} \cdot [W]_{p}^{2}$.
>                   2. Set $W$ ← $W-[H]_{:,p}^{-1}\frac{1}{[H^{-1}]{pp}}\cdot [W]{p}$.
>                   3. Set $A$ ← $[H]^{-1}_{:,p}$
>                   4. Set $B$ ← $[H]^{-1}_{p,:}$
>                   5. Set $H^{-1}$ ← $H^{-1}-\frac{1}{[H^{-1}]{pp}}AB$
>                   6. Remove $[W]_{p}$ from $W$
>
>           ---
>
>           - _Adaptive Update_:
>
>               1. Set $Y$ ← $WX$.
>               2. Update $X$ with post-process($Y$) for next layer
>
>       **To clarify, we present more notations used to calculate complexity below**
>
>       - $\( s \)$ as the sparsity ratio (a value between 0 and 1).
>       - $\( d_{in} \)$ as the input dimension of the linear layer.
>       - $\( d_{out} \)$ as the output dimension (which will align with the weight matrix's other dimension).
>       - $\( d \)$ as $\( d_{in} \times d_{out} \)$, representing the overall size of the weight matrix.
>       - $\( n \)$ as the batch size.
>       - $\( seq \)$ as the sequence length.
>
>       **Let's break down the computational complexity of each step again:**
>        1. **Adaptive Update**:
>           - Matrix multiplication $(X_{i}X_{i}^{T})$: Given $(X_{i})$ is of shape $(n \times seq, d_{in})$, and $(X_{i}^{T})$ will be $(d_{in}, seq \times n)$, resulting in a matrix of shape $(d_{in} \times d_{in})$. The complexity is $(O(n \times seq \times d_{in}^2))$.
>           - Inverting the matrix: $(O(d_{in}^3))$.
>           - $(H_{i}^{-1}X_{i}^{T}Y_{i})$: $(O(n \times seq \times d_{in} \times d_{out}))$.
>
>       2. **Pruning with Hessian Matrix**:
>           - The outer loop runs for $(d_{in})$ iterations. For each row of $(W_{i})$, the inner loop, determined by $( k = int(d_{in} \times s) )$, runs.
>               - Finding the argmin: $(O(d_{in}))$.
>               - Updating $(W_{i})$: $(O(d_{in} \times d_{out}))$.
>               - Computing tmp: $(O(d_{in} \times d_{out}))$.
>               - Updating $(H_{i}^{-1}): (O(d_{in} \times d_{out}))$.
>               - Removing row from $(W_{i}): (O(d_{in} \times d_{out}))$.
>
>           The complexity for the inner loop is $(O(k \times d_{in} \times d_{out}))$. The combined complexity for the entire pruning phase becomes $(O(d_{in}^2 \times s \times d_{in} \times d_{out}) = O(d_{in}^3 \times s \times d_{out}))$.
>
>       3. **Adaptive Update (2)**:
>           - $\(Y_{i} = W_{i}X_{i}\)$: This is matrix multiplication with complexity $\(O(n \times seq \times d_{in} \times d_{out})\)$.
>
>       4. **Combining all calculations for one layer:**
>             - $\[O(2n \times seq \times d_{in} \times d_{out} + n \times seq \times d_{in}^2 + 2d_{in}^3 + d_{in}^3 \times s \times d_{out})\]$
>
>       5. **Dominant terms:**
>             - $\[O(d_{in}^3 \times d_{out})\]$
>
>       **Based on the above analysis, the complexity of pruning one layer is:** $[O(d_{in}^3 \times d_{out})]$
>
>       > **NOTE**: The complexity of the algorithm is also substantiated by the paper `(Frantar and Alistarh; 2022)` available at: https://arxiv.org/pdf/2208.11580.pdf**)
>
>       **Key observations:**
>       - **This complexity is independent of batch size** $(n)$, because we use calibration data, which restricts (n) to small values like 128 ~ 1024.
>       - **The major factor driving the complexity is the cubic relationship with** $(d_{in})$, which for larger values of $(d_{in})$ can become significant.
>
>       **Let's compare the computational complexity with traditional layer retraining:**
>
>       For training one layer, the complexity is approximately: $[O(n \times seq \times d_{in} \times d_{out})]$
>
>       **Key observations:**
>       - **Here, the complexity is directly proportional to batch size** $(n)$, which becomes massive due to large datasets during training. This means that as the dataset size grows, the training complexity scales accordingly.
>        - While the pruning method's complexity is invariant with respect to (n), training complexity grows, making it computationally demanding for larger datasets or longer sequences.
>
>       **Comparative Insights:**
>
>       - **Batch Size \(n\)**:  Our pruning method leverages calibration data, limiting $(n)$ to smaller values (128-1024). This starkly contrasts traditional training, where $(n)$ can be considerably larger due to extensive datasets, amplifying its computational demands.
>
>       - **Dimensionality Dependency**: The pruning algorithm has a cubic dependency on $(d_{in})$, which can make it computationally intensive, especially for layers with a large $(d_{in})$. On the other hand, traditional training has a linear relationship with both $(d_{in})$ and $(d_{out})$.
>
>       - **Aggregate Computational Load**: Considering the entire training process involving multiple passes (epochs) over the data, the cumulative computational requirement can be vast. Yet, it's essential to emphasize that despite being executed once, our pruning method has a hefty computational load, especially for specific layers.
>
>       **In conclusion**, while our pruning method's computational demands are undeniably hefty, particularly for layers with a large $(d_{in})$, it's imperative to recognize the considerable computational burden posed by traditional training due to its scaling with the extensive dataset size. We believe that understanding this trade-off is crucial when considering the suitability and efficiency of our pruning approach vs. traditional retraining.
>
>     - **`Macro Perspective - Time`**: While our method does exhibit a **layerwise dependency**, which could potentially increase processing time, it's important to note that the number of layers in language models is limited. This means the time to complete the pruning process can be easily predicted with promised results as a reward.
>
> ## Question 4:
> Could you clarify the use of data points in the "Impact of Calibration Data" section?
>
> ---
>
> Thank you for inquiring about the use of data points in the "Impact of Calibration Data" section. I'd like to present a more structured explanation to address your question:
>
> 1. **Role of Calibration Data, $\(X\)$**:
>
>    The calibration data $\(X\)$ is pivotal for our methodology, as it directly influences the computation of the Hessian Matrix. As described in our work, the Hessian $\(H\)$ can be derived from $\(X\)$ as $\(H = X^TX\)$. For the context of language models, $\(X\)$ is structured with dimensions $\( (N, S, D) \)$. When computing \(H\), we reshape this data to a dimension of $\( (NS, D) \)$, ensuring it is well-suited for our mathematical operations.
>
> >   An integral part of our methodology is derived from the insights of **\citet{frantar2022optimal}**. The research underscores that Hessian values across different rows of a weight matrix are independent. This independence facilitates our optimization, where the removal of one data point predominantly affects its respective row value. This crucial property allows us to simplify our Hessian matrix computation, obtaining it directly as $\(H = X^TX\)$. Thus, the calibration data $\(X\)$ isn't just a dataset; it directly shapes the Hessian matrix, which in turn governs our pruning process.
>
> 2. **Significance in Pruning Algorithm**:
>
>    Referring to the main algorithm provided, the calibration data $\(X\)$ underpins the adaptive update step. Within this step, the inverse Hessian $\(H^{-1}\)$ is computed as $\(H^{-1} \gets (X_iX_i^T)^{-1}\)$. This relationship is fundamental because the precise calculation of the Hessian matrix ensures the effectiveness of our pruning process.
>
> 3. **Exploration of Calibration Data Size**:
>
>    In the "Impact of Calibration Data" section, our intention was to scrutinize how different sizes (i.e., varying $\(N\)$) of the calibration dataset influence our pruning strategy. Our experiments, as illustrated in Figure 3, conclusively showed an increased robustness and accuracy of sparse language models with a rising number of data points. However, there's a saturation point — a threshold beyond which adding more data ceases to enhance the model's performance.
>
> To conclude, the calibration data isn't just a variable in our approach — it's foundational. By understanding its impact and optimizing its use, we ensure the effectiveness and efficiency of our pruning process. We hope this comprehensive explanation offers a clear understanding of our method's rationale and its dependence on calibration data. We'll enhance this section with a more in-depth description and practical examples.
>
> ## Feedback and Suggestions:
> ---
>
> **Reframing the introduction:** We concur that the introduction could benefit from clearer alignment with the paper's content. We have revised it to more accurately capture our focus on proposing a robust pruning method without digressing into the broader adversarial attack landscape.
>
> **Adversarial examples:** Incorporating your suggestion, we've added a section early in the paper showcasing adversarial examples. This will give readers an immediate grasp of the challenges we're addressing.
>
> **Clarification on the Hessian and Algorithm 1:** Your feedback is valuable; we've expanded sections in appendix A. The revised manuscript will provide a clearer explanation of the Hessian's relevance and will break down Algorithm 1 more intuitively.
>
> **Consistency with Figure 5:** We've rectified this inconsistency by moving Figure 5 to the main body of the paper, ensuring its placement matches the reference.

---

### Official Review · Reviewer_MLxF · 2023-08-11

**Typos Grammar Style And Presentation Improvements:** 1. For the metric, I would suggest au…
**Soundness:** 4

**Excitement:**

4: Strong: This paper deepens the understanding of some phenomenon or lowers the barriers to an existing research direction.

**Missing References:**

1. In Line 111, the authors mention that "The extensive experiments well support our statement.". Could you add some references or explanations here?
2. In Line 202, could you add some references on the "previous studies"?

**Paper Topic And Main Contributions:**

Pruning is a popular method in compressing machine learning models, as it decreases the models' size and increase the inference efficiency. Currently, the objective of pruning has been extended from "accuracy", "sparsity" to "robustness", which focuses on the models' capability in defending against the adversarial attacks. In the paper, the author proposes a method that could better balance the three objective. Specifically, it first delves into the root cause of adversarial attacks and proposes the corresponding methods for increasing the robustness of the models. then, the authors uses the weight average method to further alleviate the adversarial attacks. Various experiments across different datasets demonstrate the efficacy of the proposed method.

**Questions For The Authors:**

1. In Line 240 - 246, the authors observes that the cause of adversarial attacks in NLP is that the semantically substitution words have very distinctive embedding information with the original words. First, does this argument hold for most of the adversarial attacks? Second, even if this argument holds, why this leads to the "highly close alignment between the sparse and dense language models". I would suggest the author to provide more intuitions or supporting experimental results for this.

2. From my perspective, security-related papers would normally evaluate their defense methods against multiple attack methods. Why do you only choose TextFooler? Could you provide more experiments against other popular adversarial attacks in text classification task?

3. Regarding the setting for experiments in Table 1, could you explain why you use the weights from Weight Average to initialize most of the baseline methods?

**Reasons To Accept:**

1. Generally the author provide very rich evaluation results on the proposed method and a very broad analysis on the method.
2. The writing is easy to follow. For example, the authors have provided intuitions on the proposed method.
3. The presentation is generally good.

**Reasons To Reject:**

1. Some arguments are not supported well. See {Questions For The Authors} for details.
2. The evaluation part is not thorough enough. See {Questions For The Authors} for details.
3. Novelty Issue. I have concerns on the novelty of the paper since it seemingly just combines several interesting methods to solve the problem of "robustness under model pruning".

**Reproducibility:**

4: Could mostly reproduce the results, but there may be some variation because of sample variance or minor variations in their interpretation of the protocol or method.

**Reviewer Confidence:**

4: Quite sure. I tried to check the important points carefully. It's unlikely, though conceivable, that I missed something that should affect my ratings.

---

> ### Author Rebuttal · Authors · 2023-08-26
>
> We sincerely thank the reviewer for their insightful comments and questions, which have offered us a clear perspective on areas for improvement. Please see our responses below to clarify the main concerns. Also, if you feel that your original concerns have been resolved, we would appreciate it if you would update your evaluation to reflect this. Thank you!
>
> ## Novelty Concern
>
> Novelty Issue. I have concerns about the novelty of the paper since it seemingly just combines several interesting methods to solve the problem of "robustness under model pruning.
>
> ---
>
> Thank you for raising the question about the novelty of our work. Our paper goes beyond just a simple amalgamation of existing methods; it introduces a distinctive adaptive pruning mechanism that addresses the core challenges persistent in the domain. Here's how our approach distinguishes itself:
>
> 1. **Deeper Insight into Adversarial Attacks**: Our paper critically examines the root cause behind the susceptibility of sparse or pruned language models to adversarial attacks. Through our observations, we argue that the robustness of a sparse language model is directly proportional to its retained pre-trained knowledge. This insight is pivotal as it shifts the paradigm from merely pruning to maintaining critical semantic understanding, which is instrumental in countering adversarial threats.
>
> 2. **Adaptive Pruning Mechanism**: The main innovation lies in our Ada-Pruning approach. While traditional pruning mechanisms often introduce significant reconstruction errors, our method emphasizes minimizing disruptions to dense language models' embedding and feature space. The novelty is evident in our iterative elimination of weights and the subsequent recalibration based on the Hessian Matrix. This iterative, layer-wise consideration ensures minimal loss of pre-trained knowledge.
>
> 3. **Holistic Consideration of Model Layers**: Our approach is not siloed to individual layers. Recognizing that the error in one layer can propagate and impact subsequent layers, we introduce the adaptive updating of the Hessian Matrix. This adaptive mechanism accounts for the cumulative effect of errors, ensuring that subsequent layers are pruned with the most updated information possible.
>
> 4. **Optimal Dense Weight Calculation**: Another unique aspect is our attention to the dense weight after pruning preceding layers. We identified that using the original weight matrix in conjunction with inputs that have been altered due to accumulated pruning errors can introduce significant discrepancies. To our knowledge, our solution to recalibrate dense weights in line with updated inputs is a novel contribution that better aligns the pruned model with its dense counterpart.
>
> 5. **Cost-Effectiveness**: Our methodology is uniquely post-training, eliminating the need for rigorous retraining processes. This improves efficiency and makes our method more practical for deployment.
>
> 6. **Demonstrable Superiority**: We've backed our method with extensive experiments, which, as the results indicate, offers a compelling trade-off between accuracy, sparsity, robustness, and cost, outperforming other state-of-the-art methods.
>
> 7. **In conclusion**, while our work draws from the broader framework of pruning and robustness literature, it incorporates novel mechanisms, insights, and solutions that significantly differentiate it from existing methods. We sincerely believe that our adaptive pruning strategy provides a fresh, efficient, and more holistic solution to the age-old challenge of robustness under model pruning.
>
> ## Question 1:
>
> In Line 240 - 246, the authors observes that the cause of adversarial attacks in NLP is that the semantically substitution words have very distinctive embedding information with the original words. First, does this argument hold for most of the adversarial attacks? Second, even if this argument holds, why this leads to the "highly close alignment between the sparse and dense language models". I would suggest the author to provide more intuitions or supporting experimental results for this.
>
> ---
>
> We appreciate the opportunity to clarify the points you've highlighted, especially concerning Lines 240-246 and the alignment between sparse and dense language models in the face of adversarial attacks.
>
> 1. **Misinterpretation of the Claim**: Firstly, we'd like to address a potential misinterpretation. Our main concern is not outlining the general cause of adversarial attacks in NLP models. Our claim zeroes in on the susceptibility of sparse language models against adversarial attacks when contrasted with their dense counterparts. Specifically, we suggest that while a dense model might be robust, traditional pruning often results in a sparse model that loses this robustness.
>
> 2. **Why Close Alignment?**: Given this context, the goal of maintaining a "highly close alignment between the sparse and dense language models" is a strategy to preserve as much of the inherent robustness of the dense model in the pruned, sparse version. Through our observations, we've identified that sparse models, post-pruning, deviate considerably in their word embeddings and intermediate features from their robust, dense counterparts. Our intuition is that by closely aligning the sparse model with the robust dense models in both embedding and feature space, we can potentially retain the original model's robustness after pruning.
>
> 3. **Adversarial Samples and Embeddings**: Your question on the universality of our observation in relation to most adversarial attacks is duly noted. Our findings primarily draw from a commonly employed technique in NLP where adversarial samples are generated by substituting semantically similar words. We designed our solution with this form of adversarial attack in mind. Given your feedback, we aim to test our method against various adversarial attacks. We will update the results accordingly based on this extension of our tests described in Question 2.
>
> 4. **Summary**: We are committed to revising the manuscript to offer clearer explanations and insights on these aspects. We believe that with these adjustments, the manuscript's content and its significance will be more comprehensible. Your feedback has been invaluable in pointing us in the right direction to refine our work.
>
> ## Question 2:
>
> From my perspective, security-related papers would normally evaluate their defense methods against multiple attack methods. Why do you only choose TextFooler? Could you provide more experiments against other popular adversarial attacks in text classification tasks?
>
> ---
>
> Thank you for your feedback, especially concerning the range of adversarial attack methods we've tested our defense against. We acknowledge that comprehensive testing against various adversarial attacks is pivotal for a security-related paper.
>
> 1. **Selection of TextFooler**: Our initial choice of TextFooler was driven by its recent popularity and effectiveness in the NLP community. However, we recognize the importance of evaluating our defense method against a diverse range of adversarial attack techniques to provide a more comprehensive validation of our approach.
>
> 2. **Addition of New Experiments**: In response to your feedback, we have expanded our experiments to encompass two more popular adversarial attacks: `BERT-Attack (Li et al., 2020)` and `TextBugger (Li et al., 2018)`. By leveraging BERT, BERT-Attack ensures fluency and semantic preservation in the adversarial text it generates. Meanwhile, TextBugger employs character-level and word-level perturbations to create adversarial examples, providing another dimension of challenge for our defense method.
>
>     **We compared our method with the state-of-the-art (SOTA) methods, namely `RobustT` ([source](https://aclanthology.org/2022.acl-long.157.pdf)) and `EarlyRobust` ([source](https://aclanthology.org/2022.emnlp-main.569.pdf)). Our approach consistently demonstrated superiority in the robustness of sparse language models across various sparsity levels.**
>
>     | Method | Dataset   | Attack | Sparsity | Accracy | Accracy under attack |
>     |----------|-----------|---------------|:----------------:|:----------:|:------------:|
>     | Ours | SST2      | TextBugger   | 2x      |   88.31%   |  **50.34%**   |
>     | Ours | SST2      | TextBugger   | 4x       |  86.93% |  **49.08%**  |
>     | Ours | SST2      | TextBugger   | 8x       | 85.6% |  **48.85%**   |
>     | RobustT | SST2      | TextBugger   | 2x      |   90.5%  |  35.6%   |
>     | EarlyRobust | SST2      | TextBugger   | 2x       |  91.2  |  36..7% |
>     |----------|-----------|---------------|----------------|----------|------------|
>     | Ours | SST2      | BERT-Attack  |  2x     |  88.31% |  **51.95%**   |
>     | Ours | SST2      | BERT-Attack    |  4x       | 86.93% | **50.57%**   |
>     | Ours | SST2      | BERT-Attack   |  8x       | 85.6% |  **49.32%**  |
>     | RobustT | SST2      | BERT-Attack    | 2x       |  90.5% |  28.3%   |
>     | EarlyRobust | SST2      | BERT-Attack    | 2x     |  91.2% |  30.2% |
>     |----------|-----------|---------------|----------------|----------|------------|
>     | Ours | IMDB      | TextBugger   | 2x   | 94.2% | **58.2%**   |
>     | RobustT | IMDB      | TextBugger   | 2x     |  93.2% | 46.1%   |
>     | EarlyRobust | IMDB      | TextBugger   | 2x   | 90.7% | 48.7%   |
>     |----------|-----------|---------------|----------------|----------|------------|
>     | Ours | IMDB      | BERT-Attack    | 2x      | 94.2% | **52.1%**   |
>     | RobustT | IMDB      | BERT-Attack    | 2x    |  93.2%   |  43.1%  |
>     | EarlyRobust | IMDB      | BERT-Attack    | 2x     | 90.7% | 43.5%  |
>     |----------|-----------|---------------|----------------|----------|------------|
>     | Ours | AGNews    | TextBugger   | 2x      | 93.2% | **62.0%**  |
>     | RobustT | AGNews    | TextBugger   | 2x      | 94.8% | 44.1%   |
>     | EarlyRobust | AGNews    | TextBugger   | 2x      | 94.1% | 46.2%   |
>     |----------|-----------|---------------|----------------|----------|------------|
>     | Ours | AGNews    | BERT-Attack   | 2x     | 93.2% | **70.8%**  |
>     | RobustT | AGNews    | BERT-Attack    | 2x    | 94.8% |  36.8%   |
>     | EarlyRobust | AGNews    | BERT-Attack    |  2x    | 94.1% | 39.3%   |
>
>     > **Note**: The hyperparameter max_candidates of Bert-Attack is set to 8.
>
> Through these expanded experiments, we hope to thoroughly examine our defense method's efficacy against various adversarial attacks across different datasets and sparsity levels.
>
> ## Question 3:
> Regarding the setting for experiments in Table 1, could you explain why you use the weights from Weight Average to initialize most baseline methods?
>
> ---
>
> Thank you for your insightful observation regarding initializing our baseline methods using weights from Weight Averaging. Your inquiry allows us to elaborate on a pivotal aspect of our methodology.
>
> 1. **Challenge with Surface-Level Features**: A central realization we embraced in our research is the propensity of language models to lean on surface-level or spurious features in data. While these features may benefit certain tasks, they become a liability when defending against adversarial attacks. An inevitable question arises: If a sparse language model struggles against adversarial threats, is the failure attributed to the inherent shortcomings of the dense model's reliance on these superficial features, or is it a by-product of the pruning process itself?
>
> 2. **Our Rationale for Using Weight Averaging**: To dissociate the effects of the pruning methodology from any intrinsic vulnerabilities of the dense model, it's crucial to begin with a robust foundation. Weight Averaging serves this purpose in our research. As detailed in our manuscript `[Sec 4.2, Weight Averaging for Robust Dense Model]`, this technique synergizes the strengths of different models, reducing their individual susceptibilities to misleading surface features. By combining models trained under diverse hyperparameters and settings (as highlighted in Table 4), we aim to craft a model less influenced by surface patterns in the data. Consequently, any pruned version of this model ideally retains this inherent robustness.
>
> 3. **Why Not Other Methods?**: While alternative methods, such as adversarial training, are available, our experiments revealed that relying solely on adversarial training doesn't consistently produce a robust model. Our decision to utilize weight averaging allows us to achieve a robust, dense model with enhanced stability, ensuring a dependable foundation resistant to adversarial challenges.
>
> 4. **Delving Deeper**: Our Weight Averaging technique is methodically detailed in Algorithm 2 within the Appendix for a more detailed understanding. This algorithm ensures an optimal blend of model weights, contributing to our goal of adversarial resistance.
>
> 5. **Summary**: Our choice of Weight Averaging as an initializer is driven by our commitment to ensure our pruning methods are assessed on a consistently robust foundation. This approach assures that any observed adversarial vulnerabilities reflect the pruning process and are not remnants of latent frailties in the dense model.
>
> ## Reference Concern 1
> In Line 111, the authors mention, "The extensive experiments well support our statement.". Could you add some references or explanations here?
>
> ---
>
>
> Thank you for drawing attention to Line 111. We understand your request for clearer explanations or references supporting our statement. Here's a breakdown of our rationale:
>
> - **Meaning of "Extensive Experiments":**
>   - Our use of the term "extensive experiments" refers to a comprehensive series of tests and validations conducted to fortify our claim.
>   - These experiments are pivotal to the proof and are intricately tied to the hypotheses derived from the content presented prior to Line 111.
>
> - **Direct Correlation with Hypotheses:**
>   - We have discussed the susceptibility of sparse or pruned language models to adversarial attacks due to a lack of pretrained knowledge
>   - Our experiments are tailored to:
>     - Measure the degree of this susceptibility.
>     - Validate our assertion that without pre-trained knowledge, sparse language models treat substitute words merely as integer identifiers.
>
> - **Breadth of Experiments:**
>   - Addressing the multifaceted challenge of balancing sparsity, accuracy, robustness, and cost necessitated a wide-ranging experimental design.
>   - Our tests included:
>     - Different adversarial attack strategies.
>     - Varying degrees of pruning.
>
> In light of the above points, we hope our response clarifies the foundational role of our extensive experiments in validating our claims and hypotheses. We appreciate your feedback and will ensure a clearer representation in the revised manuscript.
>
> ## Reference Concern 2
>
> In Line 202, could you add some references to the "previous studies"?
>
> ---
>
> The study referenced in Line 202 is taken from the paper by `Frantar and Alistarh (2022)`. The details of this paper can be found at https://arxiv.org/pdf/2208.11580.pdf.
>
> ## Presentation Concern
> For the metric, I would suggest that the author use popular abbreviations. E.g., ASR for Attack Success Rate.
>
> ---
> Thank you for your insightful feedback on our paper presentation. Your suggestion to use "ASR" for "Attack Success Rate" is straightforward and will enhance the paper's readability. Given its benefits, we will integrate this amendment.
>
> **We genuinely value your feedback, as it allows us to articulate and refine our work more clearly.**

---

### Official Review · Reviewer_q9t9 · 2023-08-12

**Soundness:** 4

**Excitement:**

4: Strong: This paper deepens the understanding of some phenomenon or lowers the barriers to an existing research direction.

**Paper Topic And Main Contributions:**

This paper shows that positive correlation between robustness of language model and its knowledge coverage. It also proposes a novel robust pruning techniques tat are adaptive to knowledge retention. It trains multiple knowledge models and only average the contributing model weights, and then applies adaptive pruning method layer-wise to get the robust and sparse model.

**Reasons To Accept:**

1. Strong performance of the adversarial robustness assessment compared to other methods.
2. Clear writing, pseudo-code, and demonstration.
3. Investigated how sparsity impact the robustness is less than assumed.

**Reasons To Reject:**

1. The Hessian matrix operation part is too inefficient

**Reproducibility:**

4: Could mostly reproduce the results, but there may be some variation because of sample variance or minor variations in their interpretation of the protocol or method.

**Reviewer Confidence:**

2: Willing to defend my evaluation, but it is fairly likely that I missed some details, didn't understand some central points, or can't be sure about the novelty of the work.

---

> ### Author Rebuttal · Authors · 2023-08-24
>
> ## Question: Hessian Matrix operation part is too inefficient
>
> We are grateful for the reviewer's keen observation concerning the efficiency of the Hessian Matrix operation in our study. Please see our responses below to clarify the concern. Also, if you feel that your original concern has been resolved, we would appreciate it if you would update your evaluation to reflect this. Thank you!
>
> We recognize the importance of addressing this highlighted concern. However, Grasping the intricate balance between computational and memory complexities and their broader implications is crucial. **To provide clarity, we offer an in-depth analysis of computational complexities from both `micro` and `macro` viewpoints, contrasting it with approaches that necessitate model retraining.** Furthermore, we elucidate other pivotal design elements in our method that bolster the overall efficiency.
>
> ### 1. **Analysis of Complexity**:
>   - **`Micro Perspective - Computation`**: When considering models like Bert_base and Bert_large, the computational requirements for the Hessian Matrix of one layer do not exceed that of model retraining in most cases. To clarify it, we analyze the complexity of our method step by step based on one linear layer.
>
>     **Algorithm**: Prune a linear layer $l$ of BERT with target sparsity `s` and calibration data `X`
>
>     1. **Input:**
>         - Collect original $X$, $W$, $Y$ for $l$.
>
>     2. **Procedure:** Pruning $l$
>
>         1. Set $W$, $X$, $Y$ ← $l$
>
>         ---
>
>         - _Adaptive Update_:
>
>             1. Calculate $H^{-1}$ ← $(XX^{T})^{-1}$
>             2. Set $W$ ← $H^{-1}X^{T}Y$
>
>         ---
>
>         - _Pruning with Hessian Matrix_:
>
>             1. Set $d_{in}$ ← input dimension.
>             2. Set $k$ ← int($d_{in} \cdot s$).
>             3. **for** $j$ from 1 to $k$ (parallel in rows):
>                 1. Calculate $p$ ← $argmin_{p\in{d_{in}}}\frac{1}{[H^{-1}]{pp}} \cdot [W]_{p}^{2}$.
>                 2. Set $W$ ← $W-[H]_{:,p}^{-1}\frac{1}{[H^{-1}]{pp}}\cdot [W]{p}$.
>                 3. Set $A$ ← $[H]^{-1}_{:,p}$
>                 4. Set $B$ ← $[H]^{-1}_{p,:}$
>                 5. Set $H^{-1}$ ← $H^{-1}-\frac{1}{[H^{-1}]{pp}}AB$
>                 6. Remove $[W]_{p}$ from $W$
>
>         ---
>
>         - _Adaptive Update_:
>
>             1. Set $Y$ ← $WX$.
>             2. Update $X$ with post-process($Y$) for next layer
>
>     **To clarify, we present more notations used to calculate complexity below**
>
>     - $\( s \)$ as the sparsity ratio (a value between 0 and 1).
>     - $\( d_{in} \)$ as the input dimension of the linear layer.
>     - $\( d_{out} \)$ as the output dimension (which will align with the weight matrix's other dimension).
>     - $\( d \)$ as $\( d_{in} \times d_{out} \)$, representing the overall size of the weight matrix.
>     - $\( n \)$ as the batch size.
>     - $\( seq \)$ as the sequence length.
>
>     **Let's break down the computational complexity of each step again:**
>      1. **Adaptive Update**:
>         - Matrix multiplication $(X_{i}X_{i}^{T})$: Given $(X_{i})$ is of shape $(n \times seq, d_{in})$, and $(X_{i}^{T})$ will be $(d_{in}, seq \times n)$, resulting in a matrix of shape $(d_{in} \times d_{in})$. The complexity is $(O(n \times seq \times d_{in}^2))$.
>         - Inverting the matrix: $(O(d_{in}^3))$.
>         - $(H_{i}^{-1}X_{i}^{T}Y_{i})$: $(O(n \times seq \times d_{in} \times d_{out}))$.
>
>     2. **Pruning with Hessian Matrix**:
>         - The outer loop runs for $(d_{in})$ iterations. For each row of $(W_{i})$, the inner loop, determined by $( k = int(d_{in} \times s) )$, runs.
>             - Finding the argmin: $(O(d_{in}))$.
>             - Updating $(W_{i})$: $(O(d_{in} \times d_{out}))$.
>             - Computing tmp: $(O(d_{in} \times d_{out}))$.
>             - Updating $(H_{i}^{-1}): (O(d_{in} \times d_{out}))$.
>             - Removing row from $(W_{i}): (O(d_{in} \times d_{out}))$.
>
>         The complexity for the inner loop is $(O(k \times d_{in} \times d_{out}))$. The combined complexity for the entire pruning phase becomes $(O(d_{in}^2 \times s \times d_{in} \times d_{out}) = O(d_{in}^3 \times s \times d_{out}))$.
>
>     3. **Adaptive Update (2)**:
>         - $\(Y_{i} = W_{i}X_{i}\)$: This is matrix multiplication with complexity $\(O(n \times seq \times d_{in} \times d_{out})\)$.
>
>     4. **Combining all calculations for one layer:**
>           - $\[O(2n \times seq \times d_{in} \times d_{out} + n \times seq \times d_{in}^2 + 2d_{in}^3 + d_{in}^3 \times s \times d_{out})\]$
>
>     5. **Dominant terms:**
>           - $\[O(d_{in}^3 \times d_{out})\]$
>
>     **Based on the above analysis, the complexity of pruning one layer is:** $[O(d_{in}^3 \times d_{out})]$
>
>     > **NOTE**: The complexity of the algorithm is also substantiated by the paper `(Frantar and Alistarh; 2022)` available at: https://arxiv.org/pdf/2208.11580.pdf**)
>
>     **Key observations:**
>     - **This complexity is independent of batch size** $(n)$, because we use calibration data, which restricts (n) to small values like 128 ~ 1024.
>     - **The major factor driving the complexity is the cubic relationship with** $(d_{in})$, which for larger values of $(d_{in})$ can become significant.
>
>     **Let's compare the computational complexity with traditional layer retraining:**
>
>     For training one layer, the complexity is approximately: $[O(n \times seq \times d_{in} \times d_{out})]$
>
>     **Key observations:**
>      - **Here, the complexity is directly proportional to batch size** $(n)$, which becomes massive due to large datasets during training. This means that as the dataset size grows, the training complexity scales accordingly.
>      - While the pruning method's complexity is invariant with respect to (n), training complexity grows, making it computationally demanding for larger datasets or longer sequences.
>
>     **Comparative Insights:**
>
>     - **Batch Size \(n\)**:  Our pruning method leverages calibration data, limiting $(n)$ to smaller values (128-1024). This starkly contrasts traditional training, where $(n)$ can be considerably larger due to extensive datasets, amplifying its computational demands.
>
>     - **Dimensionality Dependency**: The pruning algorithm has a cubic dependency on $(d_{in})$, which can make it computationally intensive, especially for layers with a large $(d_{in})$. On the other hand, traditional training has a linear relationship with both $(d_{in})$ and $(d_{out})$.
>
>     - **Aggregate Computational Load**: Considering the entire training process involving multiple passes (epochs) over the data, the cumulative computational requirement can be vast. Yet, it's essential to emphasize that despite being executed once, our pruning method has a hefty computational load, especially for specific layers.
>
>     **In conclusion**, while our pruning method's computational demands are undeniably hefty, particularly for layers with a large $(d_{in})$, it's imperative to recognize the considerable computational burden posed by traditional training due to its scaling with the extensive dataset size. We believe that understanding this trade-off is crucial when considering the suitability and efficiency of our pruning approach vs. traditional retraining.
>
>   - **`Macro Perspective - Time`**: While our method does exhibit a **layerwise dependency**, which could potentially increase processing time, it's important to note that the number of layers in language models is limited. This means the time to complete the pruning process can be easily predicted with promised results as a reward.
>
> ### 3. **Layer-by-Layer Computation for Resource Efficiency**:
>
> While the sum of Hessian Matrix computations of the entire language model is time-intensive, our approach uniquely addresses this by employing a layer-by-layer resolution strategy. This methodology means there's no necessity to simultaneously load the entire model into the memory of computational resources. Consequently, from a memory allocation standpoint, our pruning with the Hessian Matrix can be viewed as a resource-saving measure.
>
> ### 3. **Efficient Post-training Pruning**:
>
> A post-training pruning strategy is at the heart of our methodology. Unlike many other approaches that might require recurrent training sessions or exhaustive reiterations, ours stands out in its ability to save significant resources by strategically avoiding these processes.
>
> ### 4. **Justifying Computational Commitment**:
>
> While it's acknowledged that pruning with the Hessian Matrix does possess computational time costs, it's paramount to understand our larger vision. The ultimate objective isn't merely to save time but to sculpt a model characterized by three pillars: sparsity, robustness, and high performance. Such a model offers considerable advantages in real-world scenarios, particularly operational ones. Thus, the computational expenses encountered in the training phase can be viewed less as costs and more as strategic investments. Investments we're confident will provide tangible returns when our model is deployed in actual applications.
>
> ### 5. **Objective and Potential Method:**
>
> Drawing from established pruning methods utilizing the Hessian Matrix, we have pioneered an adaptive approach that stands out from the rest. Instead of merely focusing on the traditional sparsity-accuracy trade-off, our method delves deep into a holistic consideration of four pivotal dimensions: sparsity, accuracy, robustness, and pruning cost. **While we acknowledge the inherent time-consuming nature of Hessian Matrix computation, it's essential to note that our approach does not introduce this challenge.** Nonetheless, we are actively exploring advanced or approximate methods to expedite this process further. As a case in point, we are considering the utilization of an approximation function to concurrently estimate the inverse of the Hessian Matrix for all rows of weight, a technique underscored in the recent paper by `Elias Frantar (2023)` [https://arxiv.org/pdf/2301.00774.pdf].
>
> ### 6. **Conclusion**:
>
> In response to the reviewer's initial concerns, we emphasize a nuanced understanding of algorithm complexities, memory efficiency, resource-saving, trade-offs, and possible solutions. While inherent costs are tied to our approach, we believe they are offset by its many benefits and the broader objective it supports. Viewed holistically, our method offers a balanced interplay between computational demands and the primary goals of model robustness, efficiency, and real-world relevance.

---

### Meta-Review · Area_Chair_hD37 · 2023-09-08

**Recommendation:** 5

**Metareview:**

Overall, The reviewers are very positive about the soundness and excitement of this work. Reviewers highlight the strong experimental results in particular regarding adversarial attacks, the good presentation, and the extensive experimental setting. As such, I recommend acceptance the main conference.

---

### Decision · Program_Chairs · 2023-10-07

**Decision:**

Accept-Main

**Comment:**

Overall, The reviewers are very positive about the soundness and excitement of this work. Reviewers highlight the strong experimental results in particular regarding adversarial attacks, the good presentation, and the extensive experimental setting. As such, I recommend acceptance the main conference.